# Basal actomyosin pulses expand epithelium coordinating cell flattening and tissue elongation

Shun Li[1,2,6], Zong-Yuan Liu[3,6], Hao Li[2], Sijia Zhou[2], Jiaying Liu[2], Ningwei Sun[2], Kai-Fu Yang [4], Vanessa Dougados[2], Thomas Mangeat[2], Karine Belguise[2], Xi-Qiao Feng [3] ✉, Yiyao Liu [1,5] ✉ & Xiaobo Wang [2] ✉

Actomyosin networks constrict cell area and junctions to alter cell and tissue shape. However, during cell expansion under mechanical stress, actomyosin networks are strengthened and polarized to relax stress. Thus, cells face a conflicting situation between the enhanced actomyosin contractile properties and the expansion behaviour of the cell or tissue. To address this paradoxical situation, we study late *Drosophila* oogenesis and reveal an unusual epithelial expansion wave behaviour. Mechanistically, Rac1 and Rho1 integrate basal pulsatile actomyosin networks with ruffles and focal adhesions to increase and then stabilize basal area of epithelial cells allowing their flattening and elongation. This epithelial expansion behaviour bridges cell changes to oocyte growth and extension, while oocyte growth in turn deforms the epithelium to drive cell spreading. Basal pulsatile actomyosin networks exhibit non-contractile mechanics, non-linear structures and F-actin/Myosin-II spatio-temporal signal separation, implicating unreported expanding properties. Biophysical modelling incorporating these expanding properties well simulates epithelial cell expansion waves. Our work thus highlights actomyosin expanding properties as a key mechanism driving tissue morphogenesis.

Cortical actomyosin networks are often pulsatile to exert their contractile effects that reduce cell area and junctions to alter cell shape, resulting in tissue constriction, buckling, closure, and extension[1-13]. Theoretically, cell area or junction length is inversely correlated with actomyosin networks and their contractions[4,7,9,10,14]. Consistent with this contraction concept, epithelial cell expansion during *Drosophila* wing disc development is associated with a reduction of actomyosin networks in the plane of tissue expansion[15].

Paradoxically, actomyosin networks are often enhanced or polarized at the cell surface or junctions when cells undergo isotropic or anisotropic expansion with the involvement of external mechanical forces. *Drosophila* wing disc epithelial cells under exogenous stretch preserve epithelial stability through polarization of actomyosin networks at junctions along the stretching direction[16]. *Drosophila* pupal dorsal thorax epithelium responds to morphogenetic forces by increasing apical stress fibers to scale with increasing cell apical area,

[1]Sichuan Provincial Key Laboratory for Human Disease Gene Study, Center for Medical Genetics, Sichuan Provincial People's Hospital, School of Life Science and Technology, University of Electronic Science and Technology of China, Chengdu 610054 Sichuan, P. R. China. [2]Molecular, Cellular and Developmental Biology Department (MCD), Centre de Biologie Integrative (CBI), University of Toulouse, CNRS, UPS, Toulouse, France. [3]Department of Engineering Mechanics, Institute of Biomechanics and Medical Engineering, Tsinghua University, Beijing 100084, P.R. China. [4]MOE Key Laboratory for Neuroinformation, School of Life Science and Technology, University of Electronic Science and Technology of China, Chengdu 610054 Sichuan, P. R. China. [5]TCM Regulating Metabolic Diseases Key Laboratory of Sichuan Province, Hospital of Chengdu University of Traditional Chinese Medicine, No. 39 Shi-er-qiao Road, 610072 Chengdu, Sichuan, P.R. China. [6]These authors contributed equally: Shun Li, Zong-Yuan Liu. ✉e-mail: fengxq@tsinghua.edu.cn; liuyiyao@uestc.edu.cn; xiaobo.wang@univ-tlse3.fr

limiting larger cell elongation under this endogenous mechanical stress[17]. Likewise, during *Drosophila* oogenesis, follicular epithelial cells progressively increase their basal stress fibers and basal area in response to internal tissue growth stress[13,18]. However, the polarization and enhancement of actomyosin contractile properties have an inhibitory effect on the expansion of cells and tissue structures. Thus, there is a conflict between mechanical stress-induced actomyosin networks and the expansion behavior of the cell to tissue.

To coordinate internal stress during expansion, cells need to employ some spatiotemporal control strategies. Some ways by which intracellular stress is regulated in other cell behaviors hint at potential mechanisms. For temporal control, an epithelial monolayer stretched by an external force undergoes a two-stage process that relies on actomyosin dynamics to relax the stress, first a fast relaxation following a power law, and then a slow relaxation with exponential decay[19]. For spatial control, cells both at the leading edge and within the monolayer participate in modulating compressive and tensile stress of epithelial cell sheet to initiate collective cell migration by selectively activating a subset of cells over time[20]. For individual cells, spatiotemporal regulation of contractile vs. expanding properties at the subcellular level, such as acquiring dorsal stress fibers free of non-muscle Myosin II (abbreviated as Myosin-II) components, also coordinates internal stress to promote cell expansion[21]. Currently, the exact mechanism by which morphogenetic cells handle the conflict between the enhanced actomyosin networks and the process of cell expansion is unknown yet.

To understand this paradoxical situation, we characterized the behavior of follicular epithelial cells during late *Drosophila* oogenesis, as these cells display dramatic surface area increase (within a short period around 3 h) and dense stress fibers under the internal pressure of oocyte growth[22,23]. In this work, we report a cell expansion behavior that achieves the flattening and long axis elongation of follicle cells during late *Drosophila* oogenesis. This morphogenetic system reveals that pulsatile actomyosin networks transform their conventional contractile structures into expanding structures to expand epithelial cells and tissue, which can resolve the conflict between the enhanced actomyosin networks and the process of cell expansion.

## Results

### An unusual epithelial cell expansion behavior
During stages 9–10 (S9–10) of *Drosophila* oogenesis, basal pulsatile actomyosin networks along the dorsal–ventral (D–V) axis provide corset mechanical properties to limit tissue D–V growth thereby driving tissue extension in the anterior–posterior (A–P) direction[10,11,24], but whether similar actomyosin pulses exist in later stages is unclear. We performed 3-h confocal live imaging of follicle cells (Fig. 1a) labeled with E-Cadherin::GFP and Sqh::mCherry (the *Drosophila* homolog of non-muscle Myosin-II regulatory light chain[4]). We detected the presence and switching of basal Myosin-II pulses in multiple follicle cells during S10B-12, particularly in the posterior region of the egg chamber, showing a progression from reduction to recovery (Supplementary Movie 1). Using live imaging over shorter time intervals, we noticed that basal F-actin signals exhibited similar characteristics and synchronized pulsatile dynamics to basal Myosin-II signals during this period (Supplementary Fig. 1a–d). The gradual modulation of basal actomyosin pulsations was closely associated with an unreported behavior of morphogenetic cell expansion that propagates like a P-to-A wave and consists of different phases (Fig. 1b and Supplementary Movie 2). Here, we used dynamic Myosin-II intensity to quantify basal Myosin-II pulsatile behavior (Fig. 1c), relative Myosin-II and F-actin intensity to quantify the strength of basal actomyosin networks (Fig. 1d, e), and the segmentations and distribution angles to quantify stress fiber polarity (Supplementary Fig. 1e, f), in order to show the gradual modulation of basal actomyosin pulsations during

different stages. Meanwhile, we used circularity index (whose smaller values indicate that the cell has more ruffles) to quantify cell ruffling behavior (Fig. 1f), ruffle direction to quantify where ruffles occur within the cell (Fig. 1g), the area and perimeter fold change to quantify the increase in basal cell area and perimeter (Fig. 1h), and basal area and velocity to quantify cell basal expansion and migration processes (Fig. 1i, j); and these different factors demonstrate various cell behaviors including ruffling, increasing and spreading processes. Based on quantifications of actomyosin networks and cell changes, we define this morphogenetic cell expansion as 3 phases: a ruffling phase (0.5–1 h) during which posterior follicle cells gradually lost the strength (Fig. 1b–e) and D–V polarity (Supplementary Fig. 1e, f) of basal pulsatile actomyosin networks, and formed multiple randomly oriented membrane protrusions (Fig. 1b, f, g) to dramatically increase membrane fold (Fig. 1h); an area increasing phase (abbreviated as "increasing phase", 10-20 min) during which these cells restored the strength (Fig. 1b–e) and D–V polarity (Supplementary Fig. 1e, f) of basal pulsatile actomyosin networks, reduced ruffles while retaining them in the A–P membrane (Fig. 1b, f, g), and rapidly increased basal surface area (Fig. 1h, i); a multi-pulsed collective spreading phase (abbreviated as "spreading phase", 1–2 h) during which these cells maintained pulsatile actomyosin networks (Fig. 1b–e) and A–P directed ruffles (Fig. 1b, f, g), performed P-to-A directed collective spreading (Fig. 1j), and slowly increased basal surface area (Fig. 1i). Subcellular scale analysis showed that follicle cells exhibited dramatic changes in ruffling dynamics and cell area mainly at the basal domain (Supplementary Fig. 2a–e), whereas the dynamics of cell area and spreading appeared to be synchronized from the apical to the basal domain (Supplementary Fig. 2f, g). This indicates that cell expansion may be correlated with signals controlling the properties of the basal domain. Tissue-scale analysis revealed that follicle cell expansion behavior initiated posteriorly and eventually propagated towards the anterior (Fig. 1k, l and Supplementary Fig. 2h–j): temporally, posterior follicle cells enter the ruffling phase sequentially, but simultaneously enter the increasing and spreading phases; spatially, both ruffling and spreading processes progressively expand from posterior to anterior (Supplementary Fig. 2j). Notably, ruffling changes and basal actomyosin gradual modulation were detected in posterior wave-occurring cells, but not in anterior cells before wave arrival (Supplementary Fig. 2k–o). These expansion waves showed some differences between the ventral and dorsal subregions of the tissue (discriminated by Erk reporter activity[25], in Supplementary Fig. 2h): either all ventral cells or most of dorsal cells (except of dorsal–central columnar cells) underwent spreading (Fig. 1k, l and Supplementary Fig. 2i). The P-to-A orientation of spreading waves might be due to the minimal physical constraints of the D–V distributed corset[24].

Next, we characterized how follicle cells progressively enlarge their basal domains during this expansion behavior. Based on actomyosin constriction principle[26], cell surface area increase is expected to be accompanied by reduced actomyosin networks[4,7,9,10,14]. But follicle cells globally retained their basal surface area during the ruffling phase despite a marked reduction in basal actomyosin networks (Fig. 1c–f, i). Unexpectedly, the intensity of pulsatile Myosin-II signals, near random ruffles during the ruffling phase and at medial-basal region during the spreading phase (Supplementary Fig. 3a, b), was positively correlated with the ruffling area with a delay of 1–3 min (Supplementary Fig. 3c–h); moreover, Myosin-II global intensity was also positively correlated with a total basal area during the ruffling and spreading phases (Supplementary Fig. 3i–l). In contrast, during the pre-wave phase, basal Myosin-II pulses were inversely correlated with total basal area and preceded the reduction in basal area by approximately 1 min (Supplementary Fig. 3m–p). The difference thus implicates that basal actomyosin

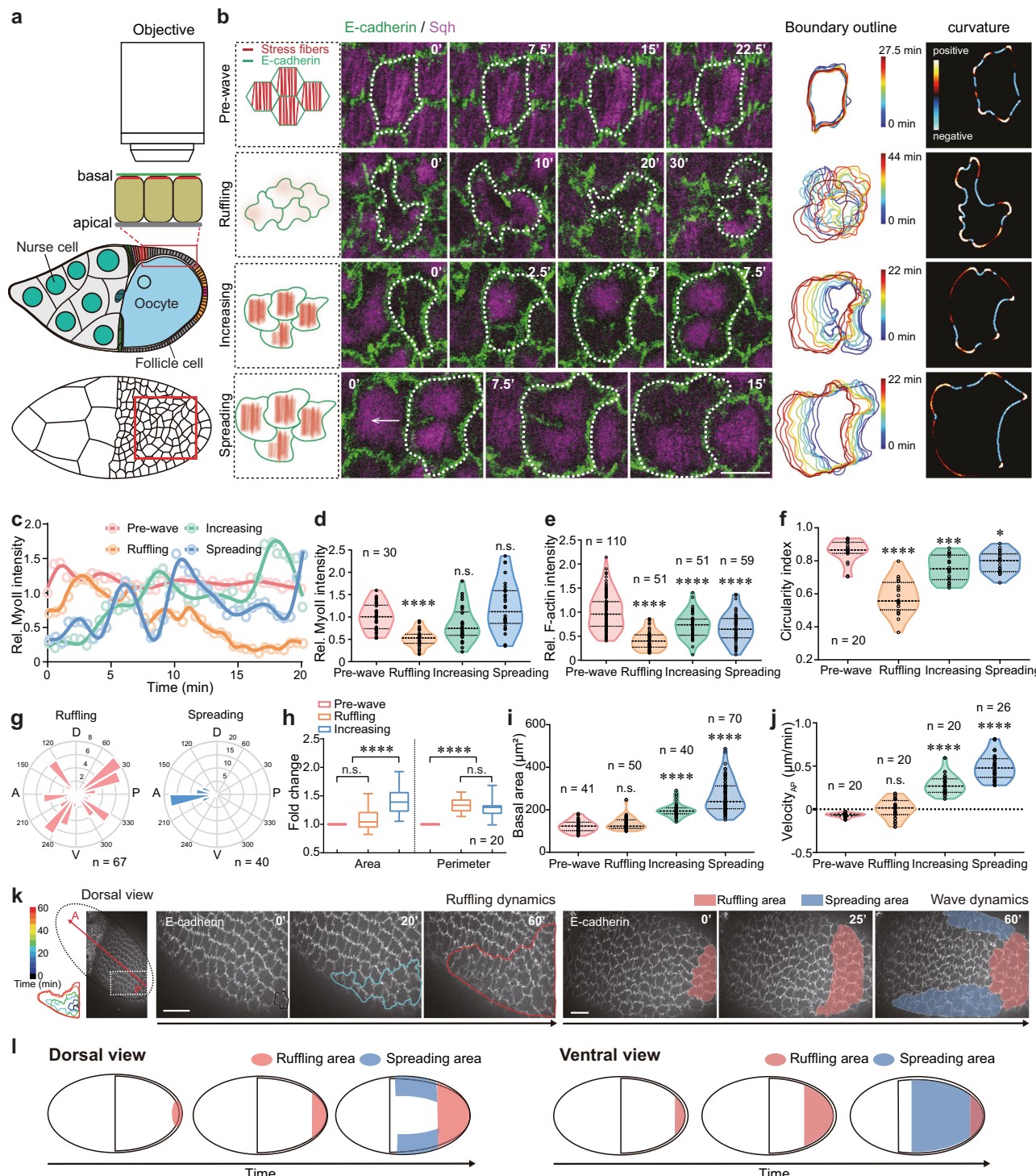

pulses during cell expansion waves may have an effect of enlarging cell area, rather than the usual constricting effect in most morphogenesis[4,7,9,10,14].

## Basal pulsatile actomyosin networks govern epithelial expansion behavior

The unusual relationship between ruffles, area, and basal actomyosin pulses led us to determine spatiotemporal signals of Rac1, Rho1, and focal adhesions. During different phases, Rac1 and Rho1 activities (respectively monitored by PAK3RBD-GFP[27,28] and AniRBD-GFP[6]) were predominantly located at follicle cell basal domains (Supplementary Fig. 4a, e), consistent with strong ruffles and pulsatile actomyosin

networks in this subcellular region. Here, pulsatile actomyosin networks are mainly basal stress fibers linked with focal adhesions at both ends (Supplementary Fig. 4j, k; ref. 13).

During the pre-wave phase, Rac1 basal activity was relatively low (Supplementary Fig. 4a, b) and was mainly distributed on the D–V junctional membrane (Supplementary Fig. 4c, d), while Rho1 basal activity was located in the medial-basal region (Supplementary Fig. 4e) which is also consistent with Myosin-II localization pattern. During the ruffling phase, Rac1 basal activity was strongly enhanced (Supplementary Fig. 4a, b) and distributed on random junctional membrane (Supplementary Fig. 4c, d), both consistent with strong random ruffles; Rho1 basal activity was significantly reduced (Supplementary Fig. 4e, f)

**Fig. 1 | Changes in basal actomyosin pulses are associated with follicle cell expansion behavior. a** Schematic representation of live cell imaging of follicle cells. **b** Representative cartoons (green and red colors mark E-cadherin and Myosin-II signals, respectively) and time-lapse images of basal domains of posterior follicle cells (labeled with E-cadherin-GFP and Sqh-RFP) in selected periods of each phase. Dotted lines mark one representative cell, and arrow marks migration direction. Boundary outlines mark the temporal changes (RGB color bars) of follicle cells. Representative images of follicle cell curvature value (reflected as color bars). **c–f** Quantification of dynamic changes of basal Myosin-II signals in one representative follicle cell (**c**), basal Myosin-II intensity (**d**) and F-actin intensity (**e**) in follicle cells, and circularity index of follicle cells (**f**). **g** Angle quantification of occurrence direction of membrane ruffles during the ruffling and spreading phases. Number at perimeter showing the angle degree, while number at radius showing the occurrence amount of membrane ruffles. D and V means the dorsal and ventral axis, respectively. **h–j** Quantifications of the increase in area and perimeter (**h**), basal surface area (**i**) and movement velocity (**j**) of posterior follicle cells. $n$ = 20 independent cells in (**h**). **k** Representative time-lapse images of follicular cell basal domains monitored by E-cadherin-GFP tracked at the tissue dorsal view, showing the initiation and the propagation of cell expansion waves. A and P represent anterior and posterior regions of egg chamber and arrow marks the wave direction. **l** Representative cartoons to summarize epithelial cell expansion waves from the dorsal and ventral views. Pink and blue colors label ruffling and spreading area, respectively, in (**k**, **l**). Scale bars are 10 μm in (**b**) and 20 μm in (**k**). The middle line shows medians, upper and lower lines as 25th and 75th percentiles, each datapoint is displayed as a dot, in (**d–f**, **i**, **j**). Boxplot shows medians, 25th and 75th percentiles as box limits, minimum and maximum values as whiskers in (**h**). The experiments in (**b**, **k**) were repeated 20 times independently. All P values have been listed in Supplementary Note 1. Source data are provided as a Source Data file.

and predominated near random ruffles (Supplementary Fig. 4g–i); consistent with Rho1 activity reduction, focal adhesions were also significantly reduced (Supplementary Fig. 4j–l and Supplementary Movie 3). During the increasing phase, Rac1 basal activity was decreased (Supplementary Fig. 4a, b) and was distributed on the A–P junctional membrane (Supplementary Fig. 4c, d); both Rho1 basal activity (Supplementary Fig. 4e, f) and focal adhesions (Supplementary Fig. 4j–l) started to recover after their strong reductions during the ruffling phase. During the spreading phase, Rac1 basal activity was partially recovered (Supplementary Fig. 4a, b) and was largely restricted to the A–P junctional membrane (Supplementary Fig. 4c, d); Rho1 basal activity was strongly enhanced (Supplementary Fig. 4e, f) and was distributed back to the medial-basal region (Supplementary Fig. 4g–i); consistent with basal Rho1 restoration, focal adhesions were strongly recovered (Supplementary Fig. 4j–l and Supplementary Movie 3). Notably, focal adhesion modulation occurred in the wave region (Supplementary Fig. 4m). Interestingly, the collective spreading process of follicle cells was a type of collective movement over the extracellular matrix [ECM] (Supplementary Fig. 4n).

Importantly, Rac1 and Rho1 basal activities, as well as focal adhesions, exhibited strong spatiotemporal correlations with ruffles, area and basal actomyosin networks. During the ruffling or spreading phase, Rac1 basal activity was positively correlated with membrane outward growth at random or anterior junctions, respectively (Supplementary Fig. 5a–c); and during the ruffling phase, pulsatile Rac1 basal activity led to a localized increase in ruffling area with a ~3-min delay, followed by basal actomyosin pulses enriched in this increased region 1–3 min later (Fig. 2a–c). During the ruffling (Fig. 2d–f) and spreading phases (Fig. 2g–i), pulsatile Rho1 basal activity occurred 0.5-1 min after localized ruffling area increase and preceded basal actomyosin pulses by 1–2 min; similarly, during the ruffing (Fig. 2j–l), increasing (Supplementary Fig. 5d–f) and spreading phases (Fig. 2m–o), focal adhesion densification also exhibited a pulsatile pattern following basal actomyosin networks. During the ruffling phase, expanding behavior of actomyosin networks and focal adhesions was unstable due to the pulsation mode (of Rac1 and Rho1 activities, and ruffling-associated local area) frequently switching between random ruffles, while during the spreading phase, this behavior was constant due to the pulsation mode always in anterior ruffles (Compare Fig. 2j with Fig. 2m). Rac1 and Rho1 activities in follicle cell basal domains appear to form an integrated signaling system to coordinate membrane ruffles, localized stress fibers and their associated focal adhesions, thereby promoting cell expansion behavior (Fig. 2p, q). It seems that when focal adhesions and their associated basal stress fibers are relatively static over ECM, pulsatile actomyosin networks might exert their contractile properties to reduce cell basal surface area[10,13], whereas when significantly expanded, pulsatile actomyosin networks might initiate newly formed focal adhesions to support transient or consecutive spreading of cell basal surface area (depending on whether ruffling is transient or persistent).

We next assessed the role of Rac1/Rho1 signaling (by temporal expression of Rac1 dominant negative [DN] form or Rho1DN in follicle cells; see Methods) in expanding follicle cells. Compared with wild-type (WT) follicle cells, genetic inhibition of Rac1 activity in follicle cells (Fig. 3a, b, f and Supplementary Movies 4, 5) caused: 1) an acute suppression of cell ruffling process (Fig. 3g), further confirmed by optogenetic Rac1 manipulation (Supplementary Fig. 6a–d; ref. 2,28,29); 2) a severe defect in basal area increase during the increasing phase (Fig. 3h), supporting ruffles as a membrane reservoir to accommodate subsequent drastic increase in basal area (Fig. 1h); 3) blockade of the cell spreading process (Fig. 3j, k), implicating the importance of ruffles for collective movement over ECM as well as its further role in increasing basal area during the spreading phase (Fig. 3i). Note that due to the limitations of genetic manipulation, e.g., the existence of genetic escapers, we observed two different levels of inhibitory effects of Rac1DN: a strong phenotype and a weak phenotype. Strong inhibition of Rac1 blocked FCs from entering the following wave process and stopped the oocyte growth. To investigate the whole process including the ruffling, increasing and spreading phases, we focused on Rac1DN weak phenotype and refer to the Rac1DN weak phenotype simply as Rac1DN in the following context. In contrast, Rho1 genetic inhibition in follicle cells (Fig. 3a, c, f and Supplementary Movies 4, 6) strongly prevented cell spreading process and its related basal area increase (Fig. 3i–k), while having no prominent effect on cell ruffling process and basal area increase before the spreading phase (Fig. 3g, h). Global photo-activation or photo-inhibition of Rho1 activity[30,31] in multiple follicle cell basal domains quickly promoted or inhibited collective cell spreading process, respectively (Supplementary Fig. 6e–g), whereas local Rho1-photoactivation in an individual cell had no prominent effect (Supplementary Fig. 6h–j). It thus not only confirms the spatiotemporal role of Rho1, but also highlights that basal pulsatile actomyosin networks present in multiple follicle cells are essential for collective spreading process. We also confirmed that the downstream effectors of Rac1 (Abi/Scar complex and its downstream Arp3 factor[28]) and Rho1 (Rock and its downstream Myosin-II[12]) are critical for cell ruffling and spreading processes (Supplementary Fig. 6k–m). Altogether, our results support that Rac1 signaling is a key factor in controlling the ruffling and spreading processes and Rho1 signaling is a key factor in controlling the spreading phase in order to expand the cell basal surface area. We also tested the role of focal adhesions by ectopic Paxillin expression or Talin RNAi expression in follicle cells[10,13]. Compared with WT follicle cells, enhanced or reduced focal adhesions in follicle cells (Fig. 3a, d–f and Supplementary Movies 4, 7, 8) significantly increased or decreased basal actomyosin networks, respectively (Supplementary Fig. 7a, b), and differently modified cell spreading process. Reduced focal adhesions in follicle cells decreased the spreading speed and disturbed P-to-A direction, thereby inhibiting basal area expansion (Fig. 3m–o). Oppositely, enhanced focal

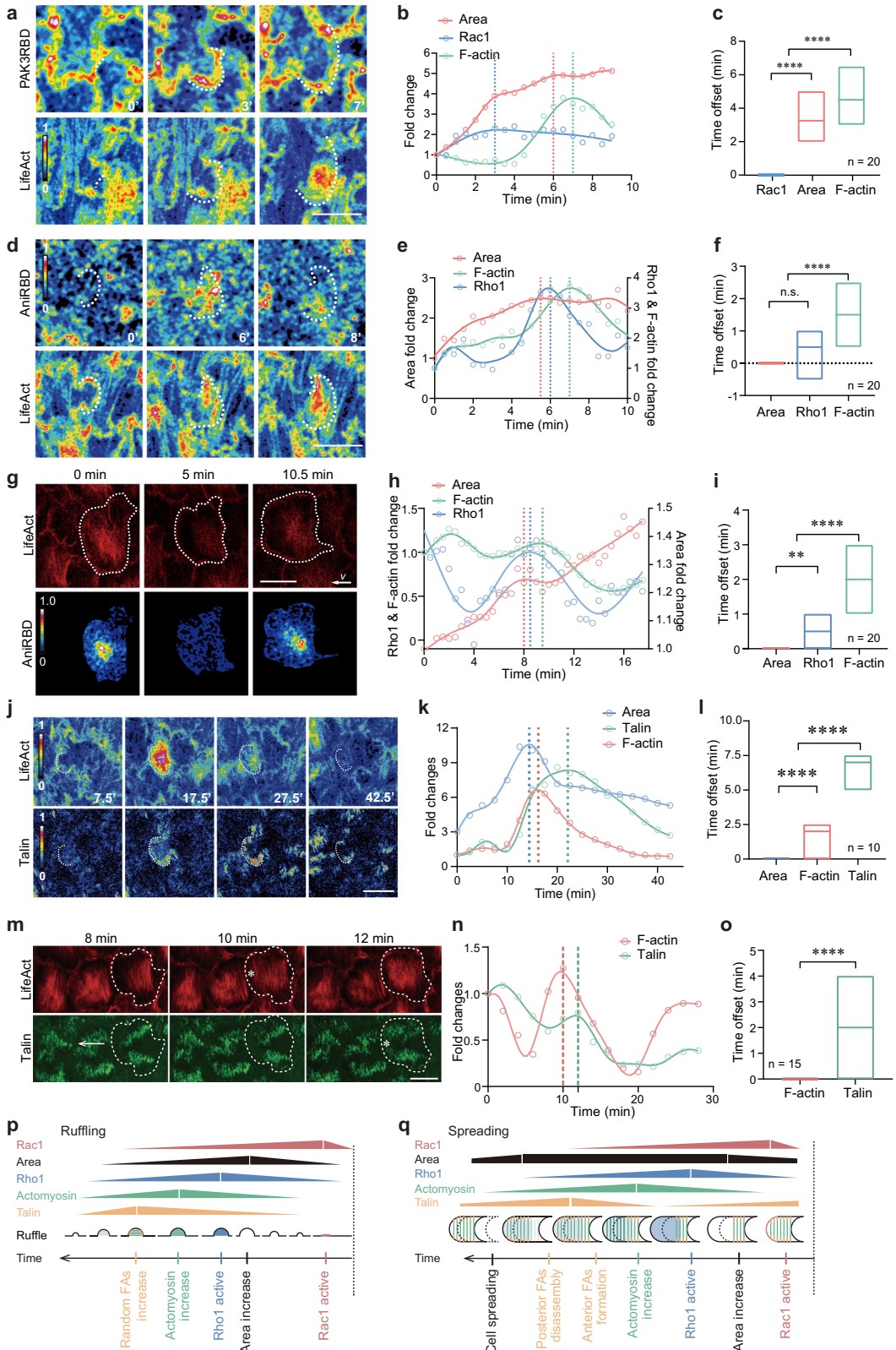

adhesions in follicle cells strongly promoted the spreading speed and basal area increase (Fig. 3m–o). Although the overall reduction in focal adhesions coincided with the onset of the ruffling process (Supplementary Fig. 4j–l), we detected a positive correlation between focal adhesions and ruffling-associated local area increase in WT follicle cells during the ruffling phase (Supplementary Fig. 7f, g). We also observed that enhanced focal adhesions in follicle

cells increased the ruffling value reflected as lower circularity index, as well as the size and timing of ruffling-associated area (Fig. 3l and Supplementary Fig. 7c–e), whereas reduced focal adhesions in follicle cells greatly decreased the size and timing of ruffling-associated area, similar to effects of Rho1 inhibition (Supplementary Fig. 7c–e). Here, dynamic ruffle formation, reflected as the circularity index, is mainly regulated by Rac1 signaling, whereas Rho1 and focal

**Fig. 2 | Follicle cell expansion behavior involves systematic changes in Rac1/ Rho1 activities and basal actomyosin networks associated with ruffles, area and focal adhesions. a, d, g, j, m** Time-lapse basal PAK3RBD-GFP and LifeAct-RFP images (heatmap) during the ruffling phase (**a**), basal AniRBD-GFP and LifeAct-RFP images (heatmap) during the ruffling (**d**) and spreading (**g**) phases, and Talin-GFP and LifeAct-RFP images (heatmap) during the ruffling (**j**) and spreading (**m**) phases. White dotted lines mark the local membrane boundary in (**a, d**) and the whole cell boundary in (**g**). White dotted lines mark a local ruffling-associated area region in (**j**) and the whole cell boundary in (**m**), arrow marks migration direction in (**m**), and asterisks show the arrival of the pulsed signals at anterior region in (**m**). RGB bars mean signal intensities in (**a, d, g, j**). **b, e, h, k, n** Quantifications of dynamic local changes of Rac1 activity, ruffling-associated subcellular basal area and F-actin signals in follicle cells during the ruffling phase (**b**), Rho1 activity, ruffling-associated basal area and F-actin signals in follicle cells during the ruffling (**e**) and spreading (**h**) phases, and basal area, Talin and F-actin signals in follicle cells during the ruffling (**k**) and spreading (**n**) phases. **c, f, i, l, o** Quantifications of the time offset of different indicated signals during the ruffling phase (**c**), the ruffling (**f**) and spreading (**i**) phases, and the ruffling (**l**) and spreading (**o**) phases. **p, q** Representative cartoons to summarize the temporal cascade composed of Rac1 activity, ruffling-associated area increase, Rho1 activity, basal actomyosin networks, and focal adhesions locally (labeled by different colors) in follicle cells during the ruffling (**p**) and spreading (**q**) phases. Scale bars are 10 μm in (**a, d, g, j, m**). The dashed lines in (**b, e, h, k, n**) indicate the peaks of corresponding color-marked signals within one cycle. The middle line indicates the median, and floating bars shows the minimum to maximum, of $n = 20, 20, 20, 10, 15$ independent cells in (**c, f, i, o**). All P values have been listed in Supplementary Note 1. Source data are provided as a Source Data file.

adhesions are primarily involved in the maintenance of ruffle timing and stability. Our results thus support an expansion role of focal adhesions and their associated stress fibers in enhancing basal surface area.

## Epithelial expansion waves bridge oocyte changes

Follicle cell expansion waves start in late S10B, overlapping with the onset of the nurse cell dumping process[32]. The fast cytoplasmic flow from the nurse cells toward the oocyte (termed dumping) leads to rapid oocyte growth and elongation along the A–P axis. The temporal coincidence of cell expansion waves and the dumping process suggests two possibilities: 1) follicle cells undergo some changes during cell expansion waves to accommodate the surface increase and A–P directed extension of the oocyte; 2) nurse cell dumping might drive cell expansion waves.

To test the first hypothesis, we characterized follicle cell shape, oocyte growth and elongation status in the WT S10B-12 egg chambers in which follicle cell number is constant[32,33]. While we detected a slight increase in follicle cell volume (Supplementary Fig. 8a), we observed a more drastic decrease in follicle cell height (Fig. 4a and Supplementary Fig. 8b), consistent with a strong increase in basal surface area (Fig. 1i), resulting in a fast and then slow flattening process during the increasing and spreading phases (Fig. 4b; a similar increase in flattening is achieved within 10–20 min or 1–2 h, thereby defining fast or slow flattening status). Furthermore, we detected follicle cell A–P extension mainly due to an increase in cell A–P length during the spreading phase (Fig. 4d and Supplementary Fig. 8c, d). Consistent with enlarged area, flattening, and elongation of follicle cells, oocyte rapidly grew mainly along the A–P axis during the spreading phase for its surface increase and A–P elongation (Fig. 4c, e, f and Supplementary Fig. 8e–g), which seems to support the first hypothesis. To further confirm this hypothesis, we tested the effect of inhibiting cell ruffling or spreading on the shape changes of the follicle cells and the oocyte. Compared with WT tissues, Rac1 genetic inhibition in follicle cells significantly delayed cell flattening thus resulting in a severe defect in oocyte growth (Fig. 4g, h and Supplementary Fig. 8h, i). This inhibition moderately repressed the cell and oocyte A–P directed extension (Fig. 4i–k and Supplementary Fig. 8j). Differently, although Rho1 genetic inhibition in follicle cells moderately slowed cell flattening, Rho1-suppressed tissues eventually grew up as WT tissues (Fig. 4g, h and Supplementary Fig. 8h, i). Strikingly, Rho1 inhibition in follicle cells strongly blocked the cell and oocyte A–P directed extension (Fig. 4i–k and Supplementary Fig. 8j). Similar to effects of Rho1 inhibition, reduced focal adhesions in follicle cells significantly decreased the oocyte A–P directed elongation (Fig. 4j, k and Supplementary Fig. 8j). Oppositely, enhanced focal adhesions in follicle cells did not prominently promote the oocyte elongated shape (Fig. 4j, k and Supplementary Fig. 8j) but fastened the elongation process (Supplementary Fig. 8k). Altogether, our results support that WT follicle cells undergo epithelial expansion waves to gradually flatten and extend along the A–P axis, in coordination with oocyte surface increase and A–P expansion (Fig. 4l).

To test the second hypothesis, we determined the effect of inhibiting Fascin in nurse cells on cell expansion waves. Fascin is a key actin-binding protein whose function in nurse cells is to control actin bundles that anchor nurse cell nuclei to prevent them from blocking the ring canal during dumping[34,35]. Egg chambers with loss-of-function (LOF) mutant of *singed* (the *Drosophila* homolog of fascin) or with *singed* RNAi expression in nurse cells were severely defective in nurse cell dumping, blocking oocyte directional growth (Fig. 5a–c). Here, we monitored cell ruffles in the anterior, middle, and posterior regions of the follicular epithelium during the pre-wave vs. ruffling phases (Fig. 5d); we then quantified the circularity index dynamics of cells in different tissue regions (Fig. 5e, g, i); finally, we acquired the temporal correlation of ruffling occurrence in different tissue regions (Fig. 5f, h, j). From this temporal correlation analysis (which showed that the decrease in the circularity index curve begins earlier in posterior cells), we confirmed that the ruffling process in WT tissues began in tissue posterior region and spread to more anterior region (Fig. 5d–f). Differently, in dumping-deficient tissues, ruffling behavior was observed in all tissue regions from anterior to posterior (Fig. 5d), indicating that the follicle cell ruffling process is not blocked by the dumping deficiency, while the circularity index curve with the same onset of decline indicated that ruffling behavior starts simultaneously throughout the tissue, losing P-to-A waves (Fig. 5g–j). Then, we characterized the spreading behavior of follicle cells by quantifying the A–P migration distance and velocity of follicle cells. Importantly, compared with WT tissues, dumping-deficient tissues strongly lost follicle cell spreading process (Fig. 5k–m), consistent with their defective oocyte growth. It thus supports that nurse cell dumping is important for driving follicle cell spreading to accommodate oocyte rapid directed growth. To note, follicle cells interact with the internal oocyte while attaching to tissue external ECM through focal adhesions. It thus indicates that P-to-A collective spreading includes two complementary modes: an active migration mode driven primarily by Rac1-induced leading protrusions[36,37], as well as a passive cell and tissue deformation due to directional oocyte growth[32,33]. This combined pattern contrasts with most collective migration models[37,38].

## Basal actomyosin networks switch mechanical properties and structures

Our results suggest that basal actomyosin networks act to expand rather than contract follicle cells during cell expansion waves. We thus implemented laser ablation to dissect the actomyosin cytoskeleton with subcellular precision to compare anisotropic tension properties at two different phases. When performing A–P directed ablation of the medial-basal network in follicle cells during the pre-wave phase, a maximum recoil speed of 0.22 μm/s was measured on average along the D–V axis (Fig. 6a, b). However, when performing laser dissection in follicle cells during the spreading phase, the medial-basal network showed little recoil along the D–V axis (a maximum recoil speed of 0.01 μm/s; Fig. 6a, b), supporting the absence of prominent tension property along the direction of fiber distribution. This absence of

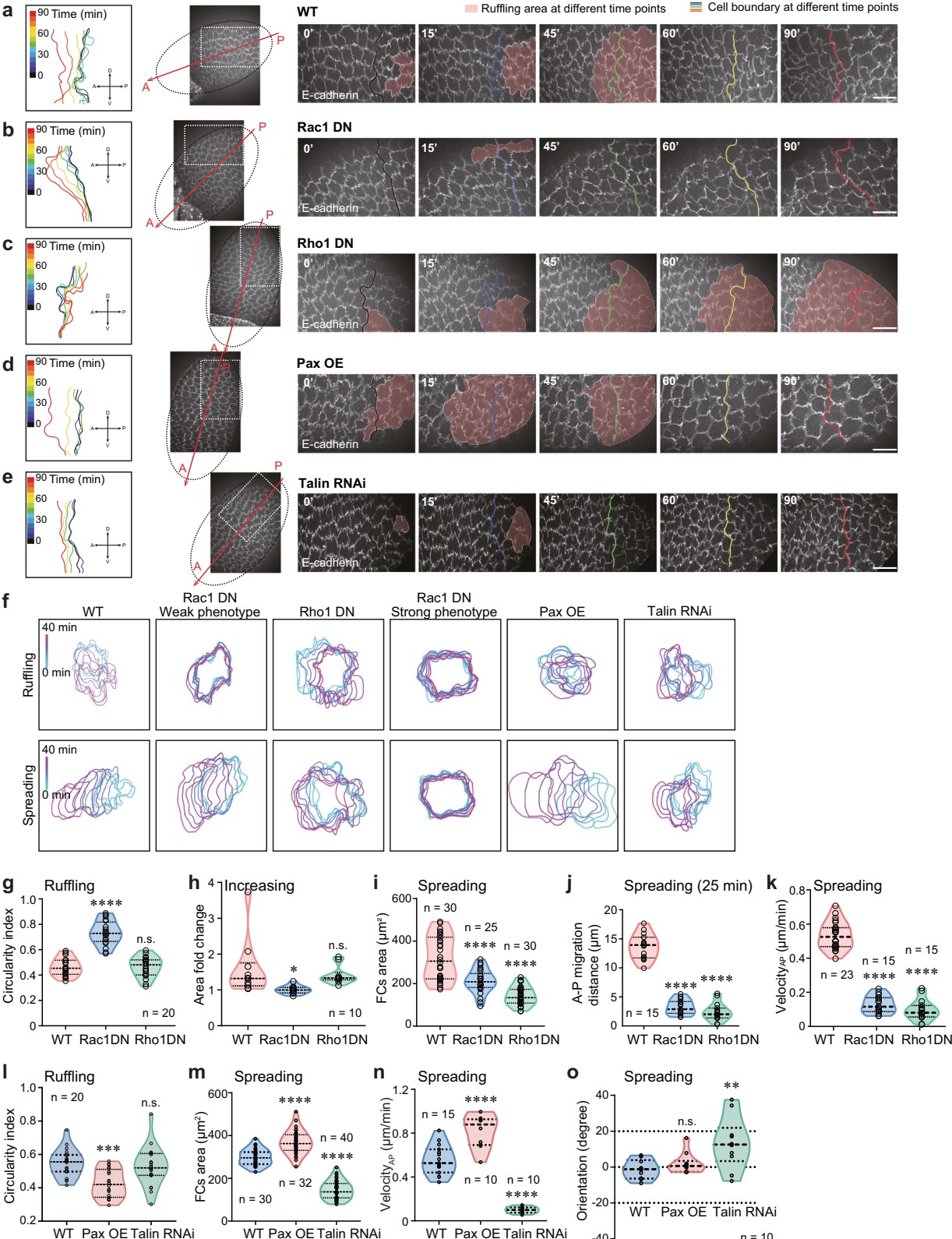

tension property during epithelial cell expansion behavior guided us to characterize fiber structures at the nanoscale.

Here, we applied random illumination microscopy (RIM[39]) to acquire super-resolved live-cell images of basal actomyosin networks at different phases, and then isolated filamentous structures from these RIM images (see Methods). During the pre-wave phase, segmented actomyosin networks exhibited a connected pattern along the D–V axis spanning almost entire cell basal domain, whereas during the spreading phase, these segmented actomyosin networks exhibited a disconnected pattern along the D–V fiber direction (Fig. 6c). Furthermore, basal actomyosin networks during the ruffling and spreading phases appeared to have a non-linear pattern consisting of weaker/ shorter filaments together with short branched structures, compared to the linear filament structures before and after cell expansion waves

**Fig. 3 | Rac1, Rho1 and focal adhesions control follicle cell expansion behavior.** **a**–**e** Representative time-lapse images of WT (**a**), Rac1DN-expressing (**b**), Rho1DN-expressing (**c**), Paxillin-overexpressing (**d**) and Talin RNAi-expressing (**e**) follicular cell basal domains monitored by E-cadherin-GFP (seen in Supplementary Movies 4–8), showing spatiotemporal effect on cell expansion waves composed of ruffling and spreading processes. A and P represent anterior and posterior regions of egg chamber and arrows mark the wave direction. Light pink colors mark ruffling area, and different colored lines mark cell boundary at different time points, in (**a**–**e**). Dotted circles mark the whole tissue boundary, and dotted rectangles mark the enlarged area showing dynamical changes of cell expansion behavior. **f** Boundary outlines mark the temporal changes (color bars) of one representative WT, Rac1DN-expressing, Rho1DN-expressing, Paxillin-overexpressing, or Talin RNAi-expressing follicle cell during the ruffing (above) and spreading (below) phases. **g**–**k** Quantifications of circularity index during the ruffling phase (**g**), basal area during the increasing phase (**h**), and basal area (**i**), migration distance (**j**) and migration velocity (**k**) during the spreading phase in the indicated follicle cells. **l**–**o** Quantifications of circularity index during the ruffling phase (**l**), basal surface area (**m**), migration velocity (**n**) and migration orientation (**o**) during the spreading phase in the indicated follicle cells. Scale bars are 20 µm in (**a**–**e**). The middle line shows medians, upper and lower lines as 25th and 75th percentiles, each datapoint is displayed as a dot, in (**g**–**o**). All P values have been listed in Supplementary Note 1. Source data are provided as a Source Data file.

(Fig. 6d, e). Next, we compared the nanoscale structure dynamics of basal actomyosin networks at different phases. Basal F-actin and Myosin-II pulsatile signals during the pre-wave phase exhibited slow velocity (mean speed 0.04 µm/min by PIV analysis), D–V contractile orientation (Fig. 6f, Supplementary Fig. 9a and Supplementary Movie 9), high spatial (Fig. 6g left panel) and temporal (Fig. 6g right panel) signal consistency (reflected as the presence of a peak; see Methods), whereas these two signals during the ruffling and spreading phases showed fast velocity (mean speed 0.2 µm/min and 0.3 µm/min, respectively, by PIV analysis), random and A–P expanding orientation during each phase (Fig. 6f, Supplementary Fig. 9a and Supplementary Movie 10), and spatiotemporal signal separation (Fig. 6g; reflected as the absence of a peak). Thus, non-linear structures and F-actin/Myosin-II spatiotemporal signal separation might explain non-contractile properties of basal pulsatile actomyosin networks during cell expansion waves. Interestingly, focal adhesion dynamics at the nanoscale exhibited either contractile or expanding behaviors during the pre-wave or spreading phase, respectively, consistent with basal Myosin-II dynamics (Supplementary Fig. 9b and Supplementary Movies 11, 12), thereby indicating that these two distinct actomyosin networks might contract or expand focal adhesions in follicle cells. Interestingly, our analysis of signal dynamics supports that the nature of basal actomyosin network dynamics is due to actomyosin filament turnover, i.e., the formation of new filaments at the cell front and the disassembly of old filaments at the cell rear; and focal adhesions also show a similar pattern to basal actomyosin networks (Supplementary Fig. 9c). Altogether, the mechanical properties as well as the nanoscale structures and their dynamics support the uncommon non-contractile yet seemingly expanding properties of actomyosin networks. It looks like that Myosin-II pulsatile signals might fluidize the cortical networks and their linked focal adhesions to execute cell expansion through stress dissipation, somehow similar to a reported role of Myosin-II on in vitro F-actin networks[40]. However, whether Myosin-II governs actin turnover or uses other unknown mechanisms to fluidize the cortical networks and enable stress dissipation is unclear.

## Theoretical modeling of the epithelial expansion suggests two migration modes

Finally, we exploited the predictive power of computational modeling (a modified cellular vertex model; Fig. 7a, b and see Methods) to theoretically simulate follicle cell expansion behavior. By incorporating our revealed expanding properties of basal actomyosin networks, our modified vertex model reproduced WT follicle cell expansion behavior in the: (1) ruffling process and P-to-A propagating waves during the ruffling phase; (2) rapid loss of ruffles during the increasing phase; (3) P-to-A collective migration with progressive increases in cell distance, basal area and velocity during the increasing and spreading phases (Fig. 7c, Supplementary Fig. 10 and Supplementary Movie 13; see Methods).

Next, we modified key variables individually (strength of Rho1, Rac1, dumping, or focal adhesions) in our simulation, allowing us to decouple the effects of a single mechanism from others. In our single-factor simulations, dumping inhibition most strongly decreased basal area expansion (Supplementary Fig. 11a, g), P-to-A migration (Supplementary Fig. 11b, f) and oocyte growth (Supplementary Fig. 11h), but did not affect the ruffling phases (Supplementary Fig. 11c–e). In addition, our single-factor simulations showed that Rac1 mainly induced cell protrusions during the ruffling phase (Supplementary Fig. 11d, e). As for the increasing and spreading phases, Rac1 induced protrusions were also the major contributor of P-to-A cell migration and expansion (Supplementary Fig. 11a, b, f–h; see Methods). However, our single-factor simulations didn't capture the strong inhibition of P-to-A cell migration and expansion as observed in experimental Rho1 inhibition (Supplementary Fig. 11a, b, f–h). To note, in single-factor simulations we pre-set identical oocyte A–P extension (also used in other genetic backgrounds), whereas in experiments Rho1DN has strongly inhibited oocyte extension (also observed in other genetic backgrounds). Thus, we propose that the difference is due to our neglect of the effect of oocyte A–P extension on cell migration, suggesting the existence of a passive migration due to oocyte A–P extension besides active migration induced by cell protrusions controlled by Rac1, Rho1, and focal adhesions. To confirm our hypothesis, we introduced different oocyte A–P extension in different genetic backgrounds as observed in experiments, which we called the combined-factor simulation (Supplementary Fig. 12 and Supplementary Movie 14; see Methods). The decline and sharp increase in the circularity index over time indicated that our combined-factor simulations reproduced the sequential occurrence of ruffling, increasing and spreading phases (Supplementary Fig. 12a); and Rac1 weak and strong inhibitions reduced the duration and dynamics of ruffles in the ruffling phase (Supplementary Fig. 12a, b), similar to the phenomena in our experiments (Supplementary Fig. 12c). Compared to the single-factor simulation method, combined-factor simulations showed that the strong inhibition of Rac 1 dramatically reduced the strength of ruffles during the ruffling phase (Supplementary Fig. 12d, e), the distance of migration (Supplementary Fig. 12f, g) and the basal area expansion (Supplementary Fig. 12h, i) during the spreading phases. In addition, our combined-factor simulations showed that the inhibition of Rho1 and the simultaneous defect in oocyte A–P extension severely hindered collective cell migration (Fig. 7d and Supplementary Fig. 12f, g), thus emphasizing that oocyte directional growth by nurse cell dumping is critical for driving passive migration mode.

## Discussion

Here, we report previously undiscovered cell expansion waves that achieve the flattening and A–P elongation of follicle cells thus accommodating rapid oocyte growth and A–P extension during late *Drosophila* oogenesis (Fig. 7e). From a theoretical view, our studies reveal a non-classical tissue morphogenetic system: nurse cell dumping induces the oocyte inflation that can initiate follicular epithelium to be flattened thereby increasing surface area; correspondingly, during the ruffling phase, follicle cells actively weaken contractile stress fibers thereby decoupling from ECM physical constraints, and concurrently with emerging ruffling behavior, follicle cells gradually increase uncommon stress fibers with expanding properties allowing easy and rapid increase in size after the ruffling phase; then, the area increasing

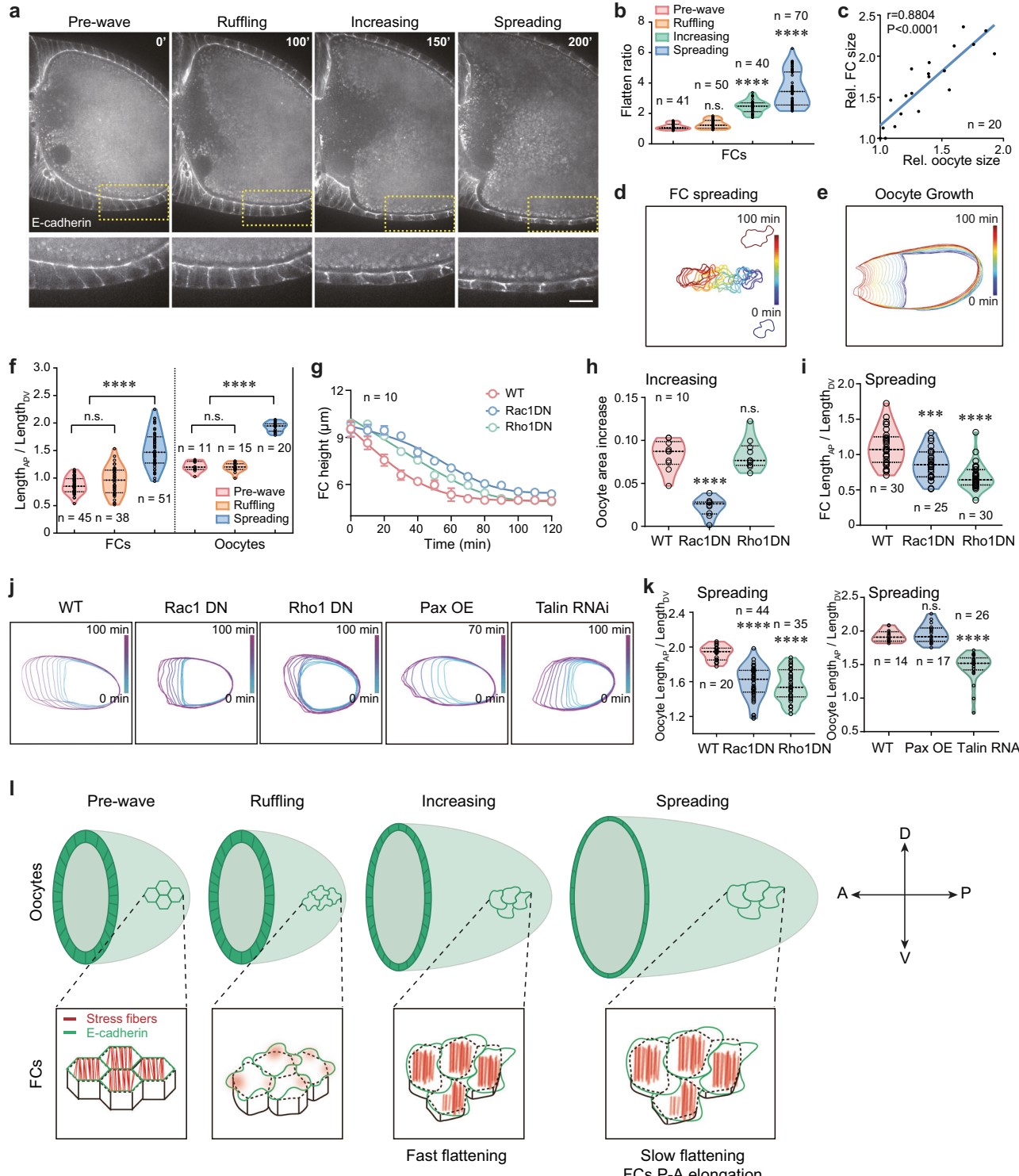

phase acts as a transition stage between the ruffling and spreading phases; finally, after the tissue overcomes a size-limiting threshold, fully restored expanding stress fibers guide the epithelium to expand preferentially in the A–P direction, thus leading to directional cell spreading and tissue elongation. Importantly, this morphogenetic system reveals that pulsatile actomyosin networks transform their conventional contractile structures into unusual expanding structures to expand epithelial cells and tissue, subverting the usual concept of actomyosin constriction effect[26].

In conventional stress fibers, Myosin-II motors can load on nearby F-actin filaments enabling local contractility, and then local

contraction forces can be integrated across interconnected networks to contract the cell. However, in expanding stress fibers, due to spatiotemporal signal separation, Myosin-II motors might not efficiently load on nearby F-actin filaments for local contractility, and meanwhile short branched networks might also limit force integration, thereby resulting in non-contractile properties. Here, how expanding stress fibers form, develop and mechanistically drive cell expansion is a crucial unanswered question. Notably, these nonclassical expanding properties of basal stress fibers do not appear to generate expanding forces that physically enlarge follicle cell basal domain; instead, activation of stress fiber actomyosin networks typically occurs following

**Fig. 4 | Follicle cell expansion behavior bridges epithelial cell flattening and elongation to oocyte growth and extension. a** Representative cross-section images of E-cadherin-GFP labeled egg chambers during the cell expansion behavior, showing follicle cell flattening process. Dotted boxes show the enlarged images (below). **b** Quantification of flatten ratio of follicle cells. **c** The correlation assay of follicle cell basal area and oocyte area in egg chambers during the spreading phase. One-sided simple linear regression, $p < 0.0001$, $R^2 = 0.7751$, $n = 20$ independent cells and oocytes. **d, e** Boundary outline marks temporal changes (RGB color bars) of one representative follicle cell (**d**) and oocyte (**e**) during the spreading phase. Since follicle cell boundary outlines from 0 to 100 min blocked us to view cell shape during the spreading process, boundary outlines at 0 and 100 min have been shown separately to highlight follicle cell elongation status. **f** Quantification of A–P length/D–V length ratio of follicle cells and oocytes. **g** Quantification of dynamic changes of the height in one representative indicated follicle cell after the ruffling phase. $n = 10$ independent cells and data are presented as mean values ± SD. **h** Quantification of oocyte area increase from the indicated follicular epithelia during the increasing phase. **i** Quantification of A–P length/D–V length ratio in the indicated follicle cells during the spreading phase. **j** Boundary outlines mark temporal changes (color bars) of one representative oocyte from WT, Rac1DN-expressing, Rho1DN-expressing, Paxillin-expressing, or Talin RNAi-expressing follicular epithelium during the spreading phase. **k** Quantifications of A–P length/D–V length ratio of the oocytes from the indicated follicular epithelia during the spreading phase. **l** Representative cartoons to summarize follicle cell expansion waves correlated with follicle cell flattening and A–P elongation as well as oocyte growth and A–P elongation. Green and red colors mark E-cadherin and stress fibers, respectively. Scale bars are 10 μm in (**a**). The middle line shows medians, upper and lower lines as 25th and 75th percentiles, each datapoint is displayed as a dot, in (**b, f, h, i, k**). All P values have been listed in Supplementary Note 1. Source data are provided as a Source Data file.

basal cell area expansion, supporting their roles in stabilizing or reinforcing basal expansion.

It is well known that focal adhesions function as "anchors" during cell migration and spreading[21]. Here, our observed expanding stress fibers demonstrate some similarities and differences with the typical structures and functions of focal adhesions during cell spreading. According to different subcellular localization and termination sites, actin stress fibers are divided into dorsal stress fibers, transverse arc and ventral stress fibers[21]. Dorsal/ventral stress fibers are anchored to the matrix surface via focal adhesions[21]. Establishing the orientation of the filaments in stress fibers is critical because it defines the contractile properties and structures, as well as focal adhesion distributions and functions[41]. In 2D cultured mesenchymal migrating cells, dorsal/ventral stress fibers show a polar distribution along the direction of cell movement; correspondingly, focal adhesions linked with dorsal or ventral stress fibers are distributed at the cell front or rear, respectively[42]. Differently, in follicle cells on the curved surface during the spreading phase, the predominant form of actin stress fibers is the ventral stress fiber perpendicular to the A–P direction of migration; correspondingly, focal adhesions are distributed on both sides of the D–V axis near the edge. Due to the different orientations of stress fibers, the direction of mature focal adhesions of mesenchymal migrating cells is elongated along the direction of cell migration[43,44], while the direction of mature focal adhesions in follicle cells is perpendicular to the A–P migration direction. Thus, the directional difference in focal adhesions and actomyosin networks leads to an adaptation of cell motility strategies. The turnover of dispersed focal adhesions in mesenchymal migrating cells requires the coordinated dynamics of 3 different stress fibers to sequentially complete the processes of focal adhesion assembly, maturation and disassembly[41]. However, in spreading follicle cells, D–V oriented actomyosin and focal adhesions assemble and disassemble at the anterior or posterior terminal of the cell, respectively, allowing cell movement in the P-to-A direction. To perform directional migration, these two types of cells use their respective strategies to establish the traction difference between the front and rear of the cell, and to establish continuous, coordinated formation and turnover of these focal adhesions, thereby anchoring or releasing cell adhesions at the cell front or rear.

Regarding follicle cell flattening during *Drosophila* late oogenesis, our study reveals multiple potential causes ranging from internal to external regions of the tissue, including: outward pressure from the expanding oocyte, adhesions between follicle cells and the elongating oocyte, and potential forces from the actomyosin networks themselves. Based on our results, we propose some unique contributions of these 3 potential causes. (1) Outward pressure from the expanding oocyte is the major driver of follicle cell flattening process, which could be indirectly reflected by the loss of follicle cell spreading behavior if dumping-induced oocyte growth is inhibited. (2) In response to oocyte outward pressure, adhesions between follicle cells

and the oocyte acts as a bridge between the increasing apical surface of follicle cells and the surface of the growing oocyte surface; correspondingly, follicle cells gradually flatten to accommodate the increasing oocyte surface. (3) During this flattening process, follicle cells gradually enlarge not only their apical surface but also their basal surface to maintain the normal epithelial cell shape; but unlike cell-cell adhesions at cell apical surface, follicle cells interact with the ECM at their basal surface[24], so the expanding properties of actomyosin networks proposed in this study could stabilize or reinforce cell basal expansion when the cell basal surface increases during cell spreading on the ECM. Here, we regard these unique contributions of different potential causes as a series of mechanical changes from internal to external tissue regions (e.g. internal pressure transfer from the oocyte to follicle cells might lead to Rac1 upregulation and Rho1 downregulation in follicle cells for their ruffling behavior); but at the same time, external mechanical responses could in turn modulate internal tissue changes[45], such as oocyte growth and elongation, indicating that E-cadherin adhesions between follicle cells and the oocyte act as a bridge mediating some feedback mechanisms to maintain structural integrity. Therefore, even with unique contributions, these potential causes are not completely separable but are highly integrated to direct follicle cell flattening and cell/oocyte changes. This interdependence has been reflected in the fact that if oocyte growth or actomyosin network expansion property is modulated, some corresponding changes occur in follicle cells or the oocyte. Thus, our observed coordination between follicle cells and the oocyte might be similar to the situation encountered during late elongation in *C. elegans*, which depends on a mechanical crosstalk between muscles and epidermis[46]. Future studies are needed to clearly understand different unique contributions as well as their interdependence for follicle cell flattening process, as well as the mechanistic feedback mechanisms of follicle cells and the oocyte to achieve their tight coordination.

Another unclear point is the unique function of Rac1 in controlling ruffling and spreading processes. Strong basal Rac1 activities, observed during the ruffling and spreading phases, are distributed at random ruffles and A–P oriented protrusions, respectively, thereby implicating different roles of Rac1 during these 2 phases. Although Rac1 is critical during the ruffling phase to drive the dynamics of random ruffles, there is no conclusive evidence that the ruffling phase defect driven by Rac1 inhibition can directly inhibit the spreading phase. Some evidence indicates that ruffling behavior of follicle cells can help the cell spreading process simple and orderly. For example, experiencing ruffling makes follicle cells plastic (more prone to morphological changes) and more susceptible to remodeling of the cytoskeleton and focal adhesions (possibly triggered by the Rho1 down-regulation in follicle cells), and also prepares the cell with sufficient membranes to allow rapid expansion. On the other hand, during cell spreading process, we observed that the first step in the pulsatile motion of follicle cells is the formation of protrusions at the anterior

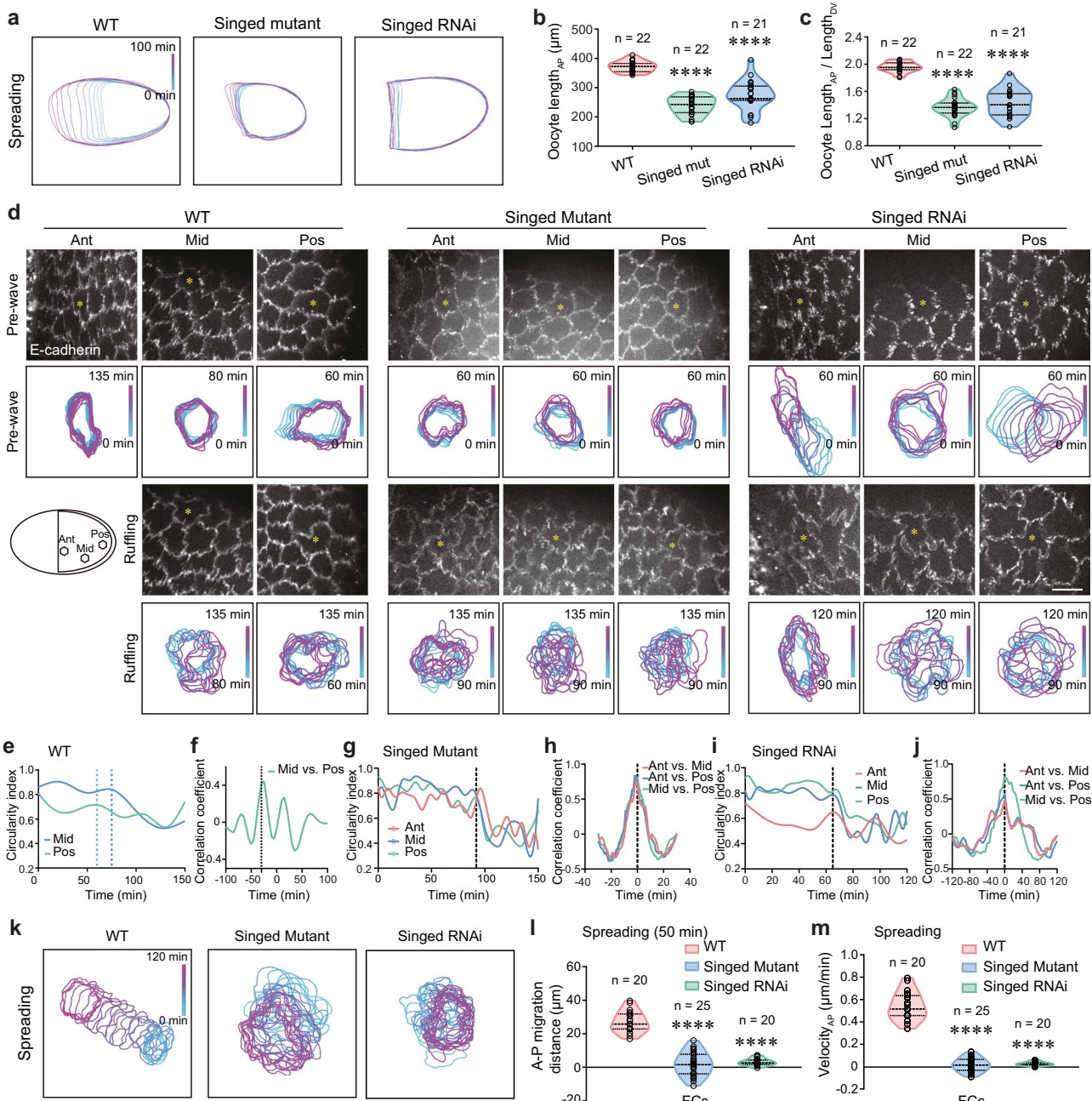

**Fig. 5 | Nurse cell dumping process drives follicle cell spreading behavior.**
**a** Boundary outlines mark the temporal changes (color bars) of one representative oocyte from WT, Singed LOF-mutant or (nurse cell specific) Singed RNAi-expressing egg chamber during the spreading phase. **b**, **c** Quantifications of A–P length (**b**) and A–P length/D–V length ratio (**c**) of the oocytes from the indicated egg chambers during the spreading phase (set by WT tissue). **d** Representative E-cadherin-GFP images (above) and boundary outlines (below) mark the temporal changes (color bars) of follicle cell basal domains in the anterior (Ant), middle (Mid) or posterior (Pos) regions of the indicated egg chambers during the ruffling phase. Yellow asterisks mark one representative cell at different tissue regions.
**e**, **g**, **i** Quantifications of dynamic changes of circularity index of follicle cells in different regions of WT (**e**), Singed LOF-mutant (**g**) or (nurse cell specific) Singed RNAi-expressing (**i**) egg chambers during the ruffling phase. The dashed lines

indicate the start time points of the ruffling process. **f**, **h**, **j** Quantifications of correlation coefficient of ruffling initiation time between follicle cells at different regions of WT (**f**), Singed LOF-mutant (**h**) or (nurse cell specific) Singed RNAi-expressing (**j**) egg chambers during the ruffling phase. The dashed lines indicate the time points of signal delay. **k** Boundary outlines mark the temporal changes (color bars) of basal domains of one representative follicle cell from the indicated egg chambers during the spreading phase. **l**, **m** Quantifications of migration distance (**l**) and migration velocity (**m**) of follicle cells in the indicated egg chambers during the spreading phase. Scale bars are 10 μm in (**d**). The middle line shows medians, upper and lower lines as 25th and 75th percentiles, each data-point is displayed as a dot, in (**b**, **c**, **l**, **m**). All P values have been listed in Supplementary Table 1. Source data are provided as a Source Data file.

edge; and signal correlation supports that these leading protrusions should be driven by Rac1. Logically, we regard the ruffling phase as a bottleneck in egg chamber development, and only follicle cells that successfully pass the ruffling can enter the subsequent process normally, as we observed in the phenotype of Rac1 strong inhibition. A

possible explanation is that follicle cells cannot decouple from the ECM without going through the ruffling phase; as the tension associated with egg chamber growth increases, the original D−V directed stress fibers during pre-wave phase will be destroyed, and the structure of epithelial monolayer and egg chamber may be compromised.

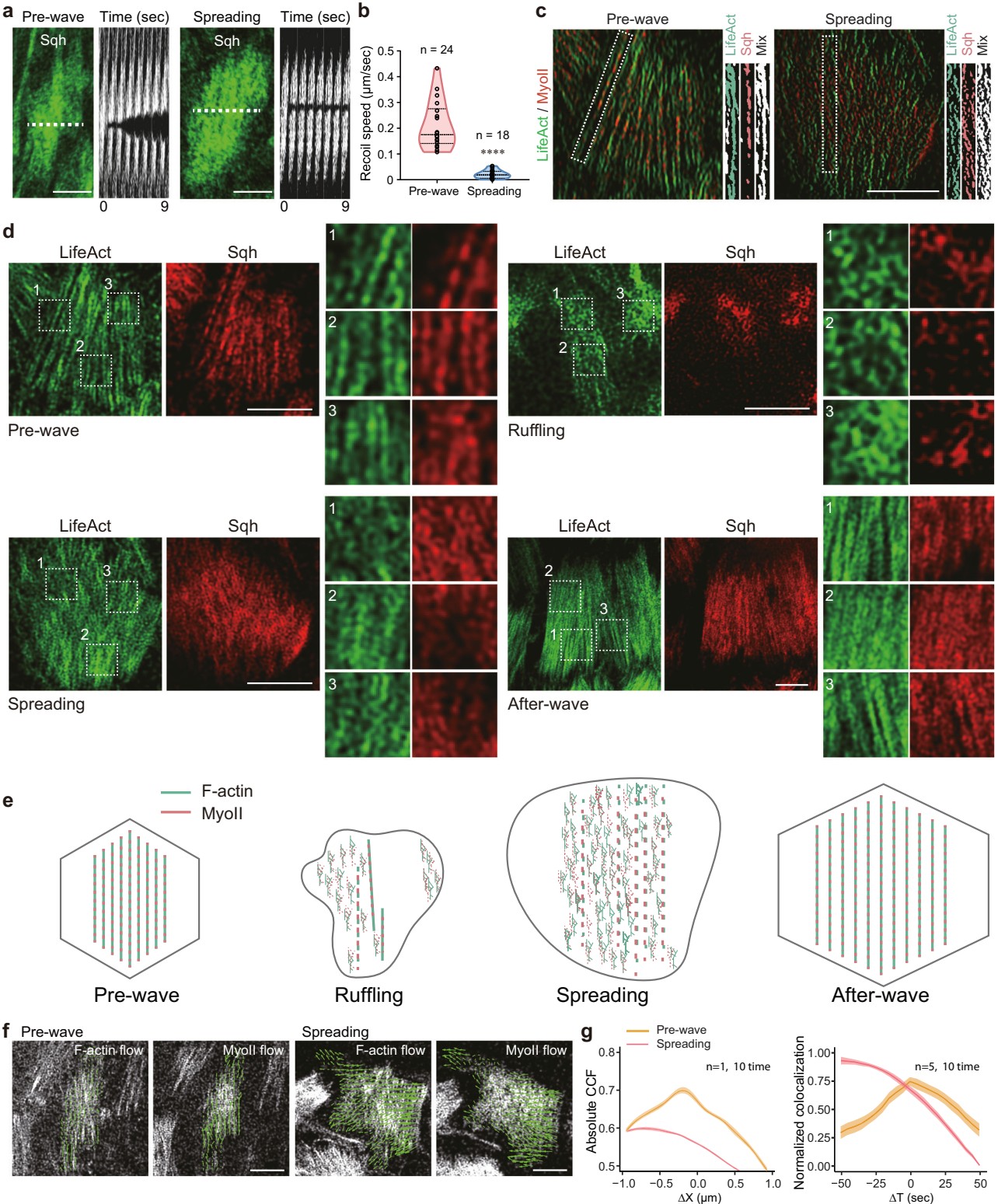

Interestingly, our study also provides some unreported aspects of mechanical adaptations related to tissue morphogenesis. Mechanical adaptations, such as stress release and mechanical redistribution, are closely related to the maintenance or disruption of homeostasis during development and pathology. Our study indicates that follicle cells might sequentially undergo two types of actomyosin-dependent mechanical adaptations during ruffling and spreading processes for follicle expansion and egg chamber elongation.

The first mechanical adaptation is membrane ruffling behavior of follicle cells. During the ruffling phase, follicle cells appear to be naturally subjected to orthogonal pressure from the underlying oocyte. Theoretically, the oocyte might experience a period of maximum pressure during the early stage of dumping, analogous to pressure changes during inflation of a balloon[47]. Here, the generation of ruffles with hyperdynamic membrane structures might be an unreported way for epithelial cells to relieve this mechanical stress. In addition to membrane ruffles, follicle cells also gradually transform

**Fig. 6 | Expanding properties of basal actomyosin networks. a** Basal Myosin-II images of a representative follicle cell before (panel on the left) and after ablation (indicated by the dashed line, represented by kymographs illustrating network recoil along the D−V axes after ablation) of basal actomyosin networks along the A−P axis, during pre-wave and spreading phases. Myosin-II signals have been visualized by Sqh-GFP. **b** Maximum recoil speed after ablations in follicle cells during the pre-wave and spreading phases. **c** Nanoscale fiber structures of basal actomyosin networks in follicle cells during the pre-wave and spreading phases. Dotted boxes mark the analysis region of F-actin and Myosin-II signal combination for the fiber connection state. The fiber signals of F-actin and Myosin-II have been combined to show either connected or disconnected filamentous structures during pre-wave or spreading phase. **d** Nanoscale structures of basal actomyosin networks in follicle cells during the indicated phases, including raw images and ROI enlarged images of F-actin and Myosin-II signals. **e** Representative cartoons summarizing linear and non-linear patterns of basal actomyosin networks before, during and after epithelial expansion behavior. Green and red colors mark F-actin and Myosin-II signals, respectively. **f** Nanoscale fiber dynamics of basal F-actin and Myosin-II signals in follicle cells during the pre-wave and spreading phases. The arrow direction indicates the signal flow orientation, and the length indicates the relative velocity. **g** Spatial (left) and temporal (right) correlation analyses of basal F-actin and Myosin-II signals (see Methods) in follicle cells during the pre-wave and spreading phases. Data are presented as mean values ± SD. Scale bars are 5 μm in (**a, c, d, f**). The middle line shows medians, upper and lower lines as 25th and 75th percentiles, each datapoint is displayed as a dot in (**b**). The experiments in (**a, c, d, f**) were repeated 10 times independently. *P* value in (**b**) have been listed in Supplementary Note 1. Source data are provided as a Source Data file.

their original contractile stress fibers into unusual expanding stress fibers, possibly further relieving this mechanical stress. Future studies are needed to test whether and how membrane ruffles and expanding stress fibers function to relieve the mechanical stress of follicle cells under the oocyte pressure. Some similar stress release has been observed in puzzle-shaped plant cells[48]. Differently, puzzle-shaped plant cells utilize microtubules to form concave sections of the cell wall within 2–3 days to resist osmotic pressure[48], whereas *Drosophila* follicle cells might apply membrane protrusions and expanding stress fibers to release cellular stress in about 1 h.

The second mechanical adaptation is the switch and coordination between the contractile and expanding effects of follicle cell actomyosin networks. In follicle cells, actomyosin contractile and expanding effects may somehow spatiotemporally separate but cooperate to control mechanical redistribution and cell shape changes. D−V polarized pulsatile stress fibers provide the physical basis of the molecular corset for oocyte A−P elongation during the pre-wave phase[24], theoretically setting the highest principal stress in the D−V direction. However, during the spreading phase, follicle cells expand along the A−P axis, instead of undergoing the classical epithelial migration pattern whose direction aligns with that of maximum principal stress[49]. Intriguing, follicle cells remodel pulsatile actomyosin networks from contractile to expanding stress fibers to dissipate the theoretically postulated D−V polarized stress, resulting in progressive A−P elongation of follicle cells. After cell expansion waves, these expanding stress fibers eventually transform back into contractile fiber structures aligned along the D−V axis. It thus suggests temporal control of stress fiber structures exerted by individual follicle cells to coordinate internal stress during epithelial expansion. At the tissue scale, pulsatile stress fibers within cells undergoing expansion waves exhibit expanding structures and properties, whereas those fibers within cells prior to the wave arrival maintain contractile structures and D−V mechanical corset properties, thereby implicating spatial control to coordinate internal stress of different cells within the epithelium. The coexistence of these two strategies seem to perfectly coordinate the contractile and expanding properties of actomyosin networks, thereby resolving the paradox between enhanced actomyosin networks and epithelial cell expansion process.

## Methods
### Drosophila stocks and genetics
The following fly stocks were used (information is listed in Supplementary Table 1): *Sqh::RLCmyosinII−mCherry* (from Eric E. Wieschaus)[4], *slbo::LifeAct−GFP*[28], *slbo::LifeAct−RFP*[28], *E-cadherin-GFP* (from Hong Yong)[50], *Talin-GFP* (from Yohanns Bellaiche)[17], *UASt-PA-RacCA*[28], *UASt-PA-RacDN*[28], *UASp-CIBN-CAAX/UASp-Cry2-RhoGEF* (Opto-RhoGEF tool from Stefano De Renzis)[31], *UASp-CIBN-CAAX/UASp-Cry2-Rho1DN* (Opto-Rho1DN tool from Bing He)[30], *Ubi::AniRBD-GFP* (from Thomas Lecuit)[6], *Singed²/FM7* (from Zennifer Zanet), *UAS-Paxillin* (from Christos G. Zervas)[13], *UAS-Rac1DN, UAS-Rho1DN, UAS-ROCKRNAi, UAS-SqhRNAi, UAS-ScarRNAi, UAS-AbiRNAi, UAS-Arp3RNAi, UAS-TalinRNAi, Talin-*

*GFP*[51]*, UAS-SingedRNAi,* and *Sqh::PAK3-RBD-GFP* (from Bloomington *Drosophila* stock center). *Tj-Gal4* was used to drive different UASt transgene including optogenetic tools in all follicle cells, except that *Nanos-Gal4* was used to drive *UAS-SingedRNAi* specifically in germ lines to produce the defect blocking nurse cell dumping process.

For genetic inhibition of Rac1 or Rho1 in follicle cells, *tubP-GAL80ts* flies are combined with *Tj-Gal4* and then crossed with *UAS-Rac1DN or UAS-Rho1DN* to prevent the leaking expression of Rac1DN or Rho1DN. All stocks and crosses were maintained at room temperature. The progeny flies from the cross between *Tj-Gal4* and *UASt-Rac1DN* or *UASt-Rho1DN* were kept at 18 °C for 1–2 days and then fattened at 29 °C for 6 h (for Rac1DN expression) or 8 h (for Rho1DN expression) before dissection. The progeny flies from the cross between *Tj-Gal4* and *UASt-RNAi* were kept at 18 °C for 1–2 days and then fattened at 29 °C for overnight (12–16 h) before dissection. The progeny flies from the cross between *Nanos-Gal4* and *UAS-SingedRNAi* were kept at 18 °C for 1–2 days and then fattened at 29 °C for overnight (12–16 h) before dissection.

For the optogenetic PA-Rac experiments, the progeny flies from the cross between *Tj-Gal4* and *UASt-PA-Rac* were kept at 18 °C for 1–2 days and then fattened at 25 °C for 6–8 h before dissection. All steps were carried on in dark conditions, including cross, maintenance, and heat shock. *Drosophila* ovaries were dissected in weak light conditions, and egg chambers were mounted under red light condition before blue light illumination.

For the optogenetic Opto-RhoGEF or Opto-Rho1DN experiments, *tubP-GAL80ts* flies are combined with *Tj-Gal4* and then crossed with *UASp-CIBN-CAAX/UASp-Cry2-RhoGEF* (Opto-RhoGEF tool) or *UASp-CIBN-CAAX/UASp-Cry2-Rho1DN* (Opto-Rho1DN) to prevent the leaking expression of either optogenetic tool. The progeny flies from the cross were kept at 18 °C for 1–2 days and then fattened at 29 °C for 2 h before dissection. All steps were carried on in dark conditions, including cross, maintenance, and heat shock. *Drosophila* ovaries were dissected in weak light conditions, and egg chambers were mounted under red light condition before blue light illumination.

### Dissection and mounting of the *Drosophila* egg chamber
One- to three-day-old females were fattened on yeast with males for 1-2 days before dissection. *Drosophila* egg chambers were dissected and mounted in live-imaging medium (Invitrogen Schneider's insect medium with 20% FBS and with a final pH adjusted to 6.9), using a similar version of the protocol described in[52]. In contrast to the normal mounting conditions, egg chambers were slightly compressed to overcome the endogenous curvature. Under this condition, basal oscillation pattern, intensity, and period were similar to those observed under conditions without compression.

### Imaging and photomanipulation
Time-lapse imaging was performed with a Leica spinningdisk confocal microscope with a 20× numerical aperture 1.3 inverted oil lens or 63x numerical aperture 1.3 inverted oil lens, with a 488 nm laser and a 568 nm laser.

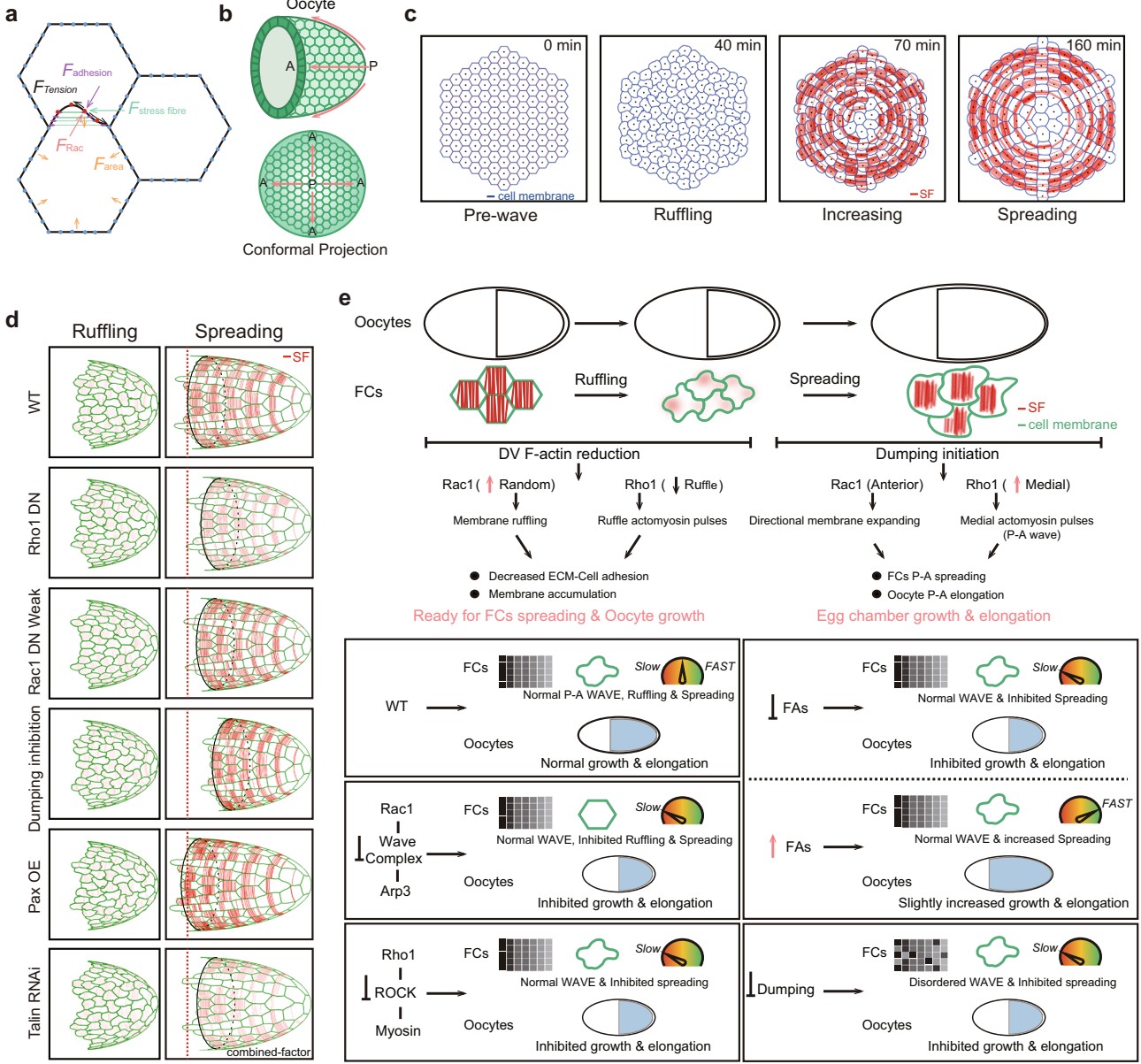

**Fig. 7 | The in silico simulation and summarization of follicle cell expansion behavior. a** Representative cartoon to summarize follicle cell basal membrane expansion behavior controlled by Rac1-mediated protrusive force, cortical tension along basal membrane, focal adhesions and their linked local stress fibers. **b** Representative cartoons for the side view and conformal projection of egg chamber during follicle cell expansion waves. A and P represent anterior and posterior, respectively. **c**, Simulation of dynamical changes of follicle cell basal domain in one WT tissue during the indicated phases. The shape of follicle cells is illustrated in blue and stress fibers along the D–V direction in the ruffling and spreading phases are illustrated in red. **d** Combined-factor simulation of oocyte elongation and follicle cell spreading behavior with the indicated genetic backgrounds during the ruffling and spreading phases. Red dotted lines mark the final position of leading follicle cells in WT tissues. In Rho1DN-expressing and Talin RNAi-expressing follicle epithelia, we introduced decreased stress fibers while in Pax-overexpressing follicle epithelium, we introduced increased stress fibers, as observed in experiments. Red color marks stress fibers in follicle cells. **e** Above: Representative cartoons to summarize the ruffling and spreading phases (here area increasing phase is unshown, considering it as a transition stage between ruffling and spreading phases), two key factors (basal Rac1 and Rho1 activities), as well as different dynamic changes in WT follicle cells and oocyte during follicle cell expansion waves. Bottom: Representative cartoons to summarize cell expansion wave (graded gray rectangles representing the propagation of expansion waves from posterior to anterior region: in most genetic backgrounds, we can detect a gradient of wave occurrence, whereas in dumping defective background, this gradient is disturbed thereby implying the loss of expansion wave), ruffling (cell shape illustrated in green) and spreading of follicle cells (cell migration speed represented by speedometer), and the growth and A–P elongation of the oocyte (blue coverage of follicle cells and egg chamber size), in different genetic backgrounds. Green and red colors mark cell membrane and stress fibers, respectively.

Time-lapse imaging with better resolution (such as 30-s interval images of PAK3RBD-GFP or AniRBD-GFP, as well as of E-cadherin-GFP, Talin-GFP or Viking-GFP, etc) was performed with a Leica SP8 pixel-scanning confocal microscope with 63x numerical aperture 1.3 inverted oil lens, with a 488 nm laser and a 568 nm laser. The basal focal plane, which is about 1 μm beneath the basal surface inside the cell, was selected during live imaging to maximize the basal Myosin-II intensity.

For the dynamics of F-actin signals, a similar basal focal plane was selected to maximize the basal F-actin intensity, as for basal Myosin-II dynamic imaging. The same microscope setup was used when comparing intensity between different samples.

For apical to basal Z-stack imaging, the Z-stack images with around 31-55 slides and 0.5 μm interval covering the whole apical to basal regions of follicle cells have been captured. The Z-stack images either around basal focal plane or across apical to basal regions have been captured every 30 s, 1 min, 2 min, or 5 min.

To capture the dynamics of egg chamber growth, the Z-stack images with 13 slides and 1.5 μm interval covering the central region of tissues have been captured, and the Z-stack images have been captured every 5 min.

Imaging data have been collected by Leica Metamorph software (version: Metamorph 7.8.13.0) or Leica SP8 LAS software (version: LAS X).

For photoactivation experiment, live-cell imaging was performed using a Leica spinningdisk confocal microscope with a 63× numerical aperture 1.3 inverted oil lens, with a 488 nm laser and a 568 nm laser. An external blue light laser (Roper system) has been integrated with this spinningdisk confocal microscope to do photoactivation experiments with a 3D mode. The external 450 nm laser was set at 35% power global control which was linked with Leica MetaMorph to allow the photoactivation by external blue light illumination.

For the photoactivation of PA-Rac at the 3D mode, 16% power from this limited global laser power was used for 0.01 ms per pixel in a 5–15 μm circle and every photoactivation illumination took approximately 1–2 s, and photoactivation illumination was carried out every 30 s.

For the photoactivation of OptoRhoGEF or OptoRho1DN at the 3D mode, due to the much higher efficiency of membrane-anchored RhoGEF or Rho1DN to activate or inhibit Rho1, 8% power from this limited global laser power was used for 0.003 ms per pixel in a 5–15 μm circle and every photoactivation illumination took approximately 0.33–0.66 s, and photoactivation illumination was carried out every 30 s for the 3D mode.

Laser-ablation experiments in *Drosophila* ovarian follicle cells were performed with a pulsed DPSS laser (λ = 532 nm, pulse length = 1.5 ns, repetition rate up to 1 kHz, 3.5 μJ/pulse) steered by a galvanometer-based laser scanning device (DPSS-532 and UGA-42, from Rapp OptoElectronic). The laser beam was focused through an oil-immersion lens of high numerical aperture (Plan-Apochromat ×63/1.4 Imm Oil or LD LCI Plan-Apochromat ×63/1.2 multi-Imm, from Zeiss) at ×0.6 zoom. Photo-disruption was produced in the focal plane. All these protocols have been described as ref. 53.

3D RIM two cameras setup was as follows: the 3D RIM home-made setup was coupled with an inverted microscope (TiE Nikon) equipped with a 100x magnification, 1.49 N.A. objective (CFI SR APO 100XH ON 1.49 NIKON) and two SCMOS camera (ORCA-Fusion, Hamamatsu) mounted in an industrial apochromatic alignment module (Abbelight SA). Fast diode lasers (Oxxius) with the wavelength centered at 488 nm (LBX-488-200-CSB) and 561 nm (LMX-561L-200-COL) were used for all experiments. The bandpass emission filters in front of the two respective cameras were FF01-514/30-25 for camera 1 and FF01-609/54-25 for camera 2. The binary phase modulator (QXGA fourth dimensions) conjugated to the image plane combined with polarization elements, was used to generate dynamic speckle on the object plane as previously described[39]. The fluorescence was collected on one or two sCMOS cameras (OrcaFlash fusion) after passing through a Stop Line quad-Notch filter (Semrock NF03-405/488/561/635E-25), two relay lenses L4 and L5 (focal equal to 200 mm), and two band pass filter (Semrock FF01-514/30-25 for GFP and RFP 330 Smerock FF01-650/92 for RFP, respectively). The synchronization of the hardware (Z-platform, cameras, microscope, laser and SLM) was driven by an upgraded version of the commercial software INSCOPER.

3D + time RIM super-resolution acquisition and reconstruction movies were performed on *Drosophilia* ovary eggs chamber expressing LifeAct-GFP and Sqh-RFP. The streaming was made with 20 mini-seconds of temporal resolution using 48 optimized random illuminations. The acquisition time of the cell was 6 s with 48 random patterns optimized for living cells for each plane. The image reconstructions were performed with the software (ALgoRIMhttps://github.com/teamRIM/tutoRIM). The Wiener filter used was 0.18, the deconvolution parameter was 0.14 and the regularization parameter was 0.15 for the images from both channels.

## Image processing and data analysis

Images were processed with MATLAB (version: R2018a) and Image J (version: 1.51j8). For all images the background (intensity of area without sample) was subtracted.

Image J was used to calculate the intensity of the signals within an individual cell or subcellular region as the average value of all pixels within the whole-cellular or subcellular area.

In the time-lapse experiments, images were processed by MATLAB to correct photo-bleaching automatically. Normally, in the absence of significant photobleaching (less than 5% decrease of intensity between two adjacent frames, especially for background signals), we did not perform a correction, but it must be ensured that under these conditions a rising signal (such as myosin-II, Rac1, Rho1 and focal adhesions) can be observed during these four phases. Once an excessive difference of intensity was detected, after confirming that it was not due to a phase conversion, we performed a correction based on two adjacent frames (especially for background signals). Although this correction is not perfect, it is sufficient to allow us to observe upward and downward fluctuations of most signals even affected by photobleaching.

Background noise was then subtracted from the corrected images. The intensity calculation was applied to generate the intensity of F-actin, Myosin, focal adhesions, PAK3RBD and AniRBD signals in follicle cells.

The distribution of oscillation periods (including F-actin, myosin, area change, focal adhesions, PAK3RBD and AniRBD signals) was generated by measuring the intervals between each pair of adjacent peaks. We applied autocorrelation to calculate the period of a time series with different time offsets. This method averages out irregularities in the sequence and gives a similar average period. To quantify the percentage distribution of the oscillating time period, the 25- to 30-min-dynamic intensity of the n individual cells from four independent egg chambers was tracked, then the oscillating cycle time of each individual cell was calculated by the auto-correlation method.

When performing a signal sequence analysis (such as of PAK3RBD, AniRBD, F-actin, basal area and focal adhesions), the delay time period showed the time interval between signal peaks, while the latency time period represented the resting period of the signal.

The follicle cell boundary was manually defined based on the image using the E-cadherin or F-actin signals as the cell membrane reference. Then, several factors were generated to quantify membrane curvature, area, perimeter, height and volume of follicle cells:

1) To represent the curvature of the cell edge, a Fiji plugin Kappa was used for curvature measurement and a customized program coded in MATLAB (MathWorks Laboratory, Natick, MA) was applied to show the heatmap of curvature on the cell edge for local ruffling status. Here, we defined the de novo detectable outward protruding structure (with positive curvature) on the cell edge as membrane ruffles.

2) The cell basal area, cell perimeter and equivdiameter (the distance between the cell center and boundary) were measured using the regionprops function in MATLAB. When studying the systematic changes in Rac1/Rho1 activities and basal actomyosin networks associated with ruffles, area and focal adhesions, the area changes refer to subcellular area associated with membrane ruffling.

3) The cell height was generated by using 3D-reconstructed Z-stacks of confocal images.

4) The volume of follicle cells was computed as $H(S_{Apical} + \sqrt{S_{Apical} \cdot S_{Basal}} + S_{Basal})/3$ where $H$ is cell height, $S_{Apical}$ is the apical cell area and $S_{Basal}$ is the basal cell area. The orientation of ruffles was analyzed by the quantification of protrusion angles; and after dividing the circumstance into 12 equal parts, the distribution of ruffle directions was shown by rose diagram.

The flatten ratio of follicle cells was computed as $\sqrt{S_{Apical} + S_{Basal}}/H$.

Some other factors were also generated to quantify the polarity of stress fibers and cell circularity value (for global ruffling status of follicle cells):

1) Stress fiber polarity was defined by the relative angle of stress fibers, which was measured using a customized program coded in MATLAB;

2) The circularity value was computed as $4\pi S_{Basal}P^{-2}$ for the quantification of circularity index to show global ruffling status, where $P$ is the cell perimeter.

Cell migration distance was generated by the total displacement distance.

Cell migration speed was quantified by velocity along either D−V or A−P axis. The cell migration velocity along either D−V or A−P direction was generated by calculating the displacement in the D−V or A−P direction per unit of time.

Angle between stress fiber and the A−P axis was measured using a customized program coded in MATLAB.

The expansion status of the follicle cells and the oocyte was quantified by the ratio between the length at the A−P axis and the length at the D−V axis. The follicle cells and the oocyte at the indicated stages were used to generate the length at the D−V axis and the A−P axis for this expansion calculation.

Particle image velocimetry (PIV) and velocity distribution of F-actin, Myosin and focal adhesion Talin signals were performed by PIVlab software with a graphical user interface (GUI) (ref. 54).

Isolation of D−V oriented fibrous signals was performed on super-resolved images of F-actin and Myosin signals by a gabor filter with the same orientation as the D−V axis. *Measurements of the maximum recoil speed* were performed with Fiji software, using the particle image velocimetry (PIV) plugin as ref. 55. For each ablation event, the velocity vector field was determined by PIV between the pre-cut frame and an image 2 s post-cut. The maximum recoil speed was estimated from the velocity vector field as for the average velocity component orthogonal to the cut line.

For RIM 2D film editing, bleaching correction was done after RIM reconstruction with open-source FIJI software (https://imagej.net/software/fiji/) based on exponential FIT from background signal. 3D drift correction FIJI plugin was performed for 3D registration.

Spatial temporal correlation analysis included spatial correlation analysis and temporal correlation analysis as follows.

**Spatial correlation analysis.** The Van Steensel cross-correlation function (VSCF) CCF was used to quantify interaction and was described previously[56]. It was obtained by calculating Pearson's coefficient after shifting the image from camera two over a distance of Δx voxels. Thresholding has been carefully adapted for each image that rejected the 10% lower intensity value for both channels. The CCF was measured with the x-shift set to 940 nm without rotation and the resulting three graphs were averaged and plotted with 95 percentiles. The Pearson coefficient for each voxel "i" was computed by

$$P = \sum \left(I_i^{D1} - I_{avg}^{D1}\right)\left(I_i^{D2} - I_{avg}^{D2}\right) / \sqrt{\sum \left(I_i^{D1} - I_{avg}^{D1}\right)\left(I_i^{D2} - I_{avg}^{D2}\right)} \quad (1)$$

where $I_{avg}^{D1}$ and $I_{avg}^{D2}$ are the averages of the camera 1 and camera 2 of the microscope. A peak was expected when there is an interaction between two proteins.

**Temporal correlation analysis.** GcoPS software, implemented in ICY, was used to measure a colocalization parameter as a function of time lag. This was a non-parametric measurement, and the software allowed the study of colocalization values as a function of time lag DT for 2D + Time images. We normalized the colocalization value as a function between 0 and 1 with the following normalization calculation:

$$P_{(0,1)} = (P - P_{\min})/(P_{\max} - P_{\min}) \quad (2)$$

A peak was expected when there is an interaction between two proteins for a specific time lag.

Box and whiskers plots (GraphPad Prism software) were used to represent the distribution of various signals: boxes extend from the 25$^{th}$ to 75$^{th}$ percentiles, the mid-line represents the median and the whiskers indicate the maximum and the minimum values.

## Mathematical modeling

In this supplementary text, we described the biophysical model that we have developed to understand the behavior of follicle cell expansion waves and some experimental validations of the models' predictions. Section I introduces the biophysical basis of our modeling (Fig. 7a, b), including different forces corresponding to the revealed mechanisms and the setting of our modeling. In section 2, we applied this modeling to theoretically simulate the behavior of follicle cell expansion waves in the WT tissues. In section 3, we performed the single-factor simulation of follicle cell expansion waves. Considering the deviation of single-factor simulation, in section 4, we performed the combined-factor simulation, and confirmed the simulated results with experimental observations.

## Modified vertex model for the follicle cell expansion behavior

A modified vertex model is generated to account for the complex cell shape changes and cortex dynamics during follicle cell expansion. E-cadherin adhesions in follicle cells are not continuous, but locally enhanced in multiple spots across the membrane linking one cell to its neighbors (as shown in Fig. 1b). These E-cadherin clusters are in alignment with previous quantification of cell membrane in epithelial cells using super-resolution microscopy[57]. Classical vertex model has limited number of vertices to describe polygonal cell shapes[58]; here, we increased the number of vertices and attached them to E-cadherin clusters (as shown in Fig. 7a), enabling us to characterize follicle cell shape changes with large deformation and high dynamics. Tissue and cellular changes were determined by the dynamics of vertices distributed along cell basal membrane (Fig. 7a), which follows the force equilibrium, namely

$$\eta_{Vis} \frac{d\mathbf{x}}{dt} = \mathbf{F}_{Tension} + \mathbf{F}_{Area} + \mathbf{F}_{Rac} + \mathbf{F}_{Adhesion} + \mathbf{F}_{Fiber} + \mathbf{F}_{Noise} \quad (3)$$

where $t$ is time, $\mathbf{x}$ is the position of vertex, and $\eta_{Vis}$ is the drag coefficient resulting from the tissue viscosity. Multiple molecular mechanisms have been considered by introducing the resulting forces on vertices: $\mathbf{F}_{Tension}$ stands for cortical tension, $\mathbf{F}_{Area}$ is area stiffness of cell resulting from internal osmatic pressure, $\mathbf{F}_{Rac}$ is the Rac1-induced protrusion force, $\mathbf{F}_{Adhesion}$ is the cell-matrix adhesion force, $\mathbf{F}_{Fiber}$ stands for the stress of stress fibers and $\mathbf{F}_{Noise}$ is the Gaussian white noise arising from thermal fluctuations. The governing equation is integrated numerically using finite difference method with a time step $\Delta t$.

For a single vertex,

$$\mathbf{F}_{\text{Tension}} = K_L \mathbf{s}^{\pm} \tag{4}$$

where $K_L$ is the line tension alone the cell cortex and $\mathbf{s}^{\pm}$ denotes the vector pointing to the neighboring vertices.

$$\mathbf{F}_{\text{Area}} = -K_a \hat{s}(A - A_0)\mathbf{r}_{\text{nor}} \tag{5}$$

where $K_a$ is the areal elastic modulus of cell, where $\hat{s} = (|\mathbf{s}^+| + |\mathbf{s}^-|)/2$ is the coverage cortical length of current vertex, $A - A_0$ is the change of cell area during morphogenesis, $\mathbf{r}_{\text{nor}}$ is the normal direction of local cell cortex.

$$\mathbf{F}_{\text{Rac}} = f_{\text{Rac}} \frac{1}{\sigma\sqrt{2\pi}} e^{-\frac{1}{2}\left[\frac{2t-T}{2\sigma}\right]} \mathbf{r}_{\text{nor}} \tag{6}$$

where $f_{\text{Rac}}$ is the normal strength of Rac protrusion, $\sigma$ is the standard deviation of Rac strength within one Rac cycle, $T$ is the duration of the activated Rac cycle and $t$ is current morphogenesis time. In the simulation, we considered two important points. First, we defined $\mathbf{F}_{\text{Rac}}$ to follow a wave of Rac1 activity which first increases and then decreases as experimentally observed (Fig. 2a–c). Second, we assumed that $\mathbf{F}_{\text{Rac}}$ spatial localization shifted from random membrane during ruffling phase to the A–P membrane during the increasing and spreading phases, also as experimentally observed (Supplementary Fig. 4c, d).

Since Rho activity controls actomyosin contraction that can stabilize cell-matrix adhesion[13,59], we adjusted the strength of $\mathbf{F}_{\text{Tension}}$, $\mathbf{F}_{\text{Fiber}}$ and $\mathbf{F}_{\text{Adhesion}}$ to depend on Rho1 activity in our simulation; and moreover, $\mathbf{F}_{\text{Adhesion}}$ depends on the strength of cell adhesion molecules (CAMs) such as Talin and Paxillin which also affects stress fiber formation and $\mathbf{F}_{\text{Fiber}}$. The adhesion force between cell and matrix depends on two molecular mechanisms connected in series: the first is the adhesion strength of cell adhesion molecules (CAMs) such as Talin and Paxillin, and the second is the RhoA GTPase activity which controls the strength of stress fiber connecting the CAMs. Here, we use a Hill Function to quantify the adhesion

$$\mathbf{F}_{\text{Adhesion}} = f_{\text{CAMs}} \frac{L_{\text{Rho}}^{\,n}}{K_L + L_{\text{Rho}}^{\,n}} \cdot \Delta\mathbf{x} \tag{7}$$

where $f_{\text{CAMs}}$ stands for the adhesion strength of CAMs and $L_{\text{Rho}}$ is the normal concentration of RhoA GTPase, $K_L$ and $n$ are the apparent dissociation constant and the Hill coefficient, respectively. In the simulation, the strength of focal adhesion also has a time delay following the Rac activation, which is in consistence with our experimental observations (Fig. 2j–o).

$$\mathbf{F}_{\text{Fiber}} = K_{\text{Fiber}}(\mathbf{L} - \mathbf{L}_0) \tag{8}$$

where $L$ is the current length of the stress fibers and $L_0$ is the critical length of stress fibers (here stress fibers are composed of both filaments and short branched structures). In the simulation, we assume that theses stress fibers can only provide stress to perform expanding properties and cannot sustain compression (considering our observed expansion effect). The direction of stress fiber is set along the D–V axis, in consistent with our experimental observations (Supplementary Fig. 1e, f and Supplementary Fig. 4j, k).

$$\mathbf{F}_{\text{Noise}} = f_R \mathbf{r}_{\text{Noise}} \tag{9}$$

where $\mathbf{r}_{\text{Noise}}$ is the unit-variance Gaussian noise vector at current vertex and $f_R$ is the strength of the Gaussian white noise. The parameters used in the simulations are adopted from previous experimental or theoretical studies in the literature[60–63]. For single-factor and combined-factor simulations, different parameters are reduced or increased accordingly: in RhoDN, $L_{\text{Rho}}$ the concentration of RhoA GTPase is reduced; in RacDN, $f_{\text{Rac}}$ the strength of Rac protrusion is reduced; in dumping inhibition, the egg chamber growth in the A–P axis is reduced; in PaxOE, $f_{\text{CAMs}}$ the adhesion strength of CAMs is increased whereas in Talin RNAi, $f_{\text{CAMs}}$ is reduced.

Nurse cell dumping was also introduced in our simulation to manipulate the oocyte growth as experimentally observed (Fig. 4e, f, h, j, k and Supplementary Fig. 8h–k). During late stages of *Drosophila* oogenesis, nurse cells rapidly transfer their cytoplasmic contents into the oocyte, thereby increasing the oocyte length along the A–P axis. As the oocyte grows along the A–P axis, follicle cells deform and migrate on the oocyte surface. In the simulation, we introduced the effects of nurse cell dumping by modulating the A–P growth rate of the oocyte.

To better capture the cell expansion waves observed on oocyte surface in both 2D and 3D, we also applied in our simulation a conformal projection previously used to characterize *Drosophila* embryo gastrulation (Fig. 7b; ref. 64).

**Theoretical simulation of the follicle cell expansion behavior in WT tissues.** We first explored whether adding these features to our modified vertex model could or not replicate cell and tissue changes after S10B. The tissue posterior region with an initial cell shape of tightly packed hexagons was simulated, with the first 60 min being the ruffling phase, and the next 100 min being the increasing and spreading phases (Fig. 7c and Supplementary Movie 13). Overall, this vertex model reproduced follicle cell expansion waves by simulating sharp changes in cell morphological circularity parameter at different tissue posterior positions (Supplementary Fig. 10a). Consistent with our experimental observations, our simulation showed: (1) dynamic membrane protrusions, decreased cell circularity and unchanged cell basal area during the ruffling phase (Fig. 7c and Supplementary Fig. 10b, c); (2) a rapid increase in cell circularity in the increasing phase (Supplementary Fig. 10b), suggesting a sharp cell geometry transformation in such a short period; 3) P-to-A collective migration with gradually increased cell distance, basal area and velocity during the increasing and spreading phases (Fig. 7c and Supplementary Fig. 10c–e). During the spreading phase, follicle cells adapted into an A–P polarized shape with residual membrane protrusions, which benefits cell migration in this direction (Supplementary Fig. 10f).

**Single-factor simulations of various backgrounds affecting follicle cell expansion behavior.** Next, we explored several experimentally identified key factors (including the spatio-temporal activity of Rac1 and Rho1, cell-matrix adhesions and dumping-mediated oocyte growth) to understand their role in follicular cell expansion waves. We applied single-factor variable (inhibition of Rho1, Rac1, dumping, or focal adhesion; or enhancement of focal adhesion) in our simulation allowing us to decouple the effects of a single mechanism from others, which is often quite challenging to achieve experimentally.

1) For the simulation effects on the ruffling phase, these five factors had almost no effect on cell basal area and distance (Supplementary Fig. 11a–c), whereas Rac1 and Rho1 inhibitions, as well as enhanced focal adhesions, significantly changing the cell circularity (Supplementary Fig. 11d, e).

2) For the simulation effects on the spreading phase, both Rac1 and dumping inhibitions significantly blocked follicle cell P-to-A migration; the Rho1 and focal adhesion inhibitions reduced cell-matrix adhesions and then retracted/damaged cell protruded membrane, leading to the moderate reduction in migration ability; conversely, focal adhesion enhancement stabilized cell A–P protruded

membrane, promoting cell migration ability (Supplementary Fig. 11b, f, g).

3) For the simulations on the oocyte changes, inhibitions of Rac1, Rho1, focal adhesions and nurse cell dumping decreased oocyte size and A–P elongation, whereas focal adhesion enhancement increased both (Supplementary Fig. 11h).

Here, our single-factor simulations assumed that collective spreading is an Rac1-induced active collective cell migration. Considering the tight relationship between follicle cells, internal oocyte and external ECM, oocyte A–P extension (mainly along the A–P axis due to external corset limitation) also resulted in passive P-to-A migration of follicle cells over the ECM, even without active cell migration. Thus, our single-factor simulations accounted for passive migration due to oocyte A–P extension and active migration regulated by Rac1, Rho1 and CAMs, while these simulations decoupling the two migration modes led to some deviations from the actual experimental observations. For example, simulation of Rho1 strong inhibition exhibited a much weaker inhibitory effect on follicle cell P-to-A migration than that observed in experiments (Supplementary Fig. 11f–h).

**Combined-factor simulations of various backgrounds affecting follicle cell expansion behavior.** To overcome these deviations, we then performed combined-factor simulations that considered experimental situations where multiple factors change together (for example, oocyte A–P extension was also affected by Rho1 or Rac1 inhibition in follicle cells). So, we introduced different oocyte A–P extensions into different conditions according to experimental observations.

The simulation time of ruffling phase in Rac1 inhibition has been set as 40 min, in accordance with the inhibition of ruffling in our experiments. In these combined-factor simulations, Rac1 strong inhibition resulted in the most dramatic loss of membrane protrusions, theoretically supporting Rac1 signaling as the key regulator of the ruffling phase; differently, both Rho1 and focal adhesion inhibitions slightly reduced membrane protrusions, whereas focal adhesion enhancement increased membrane protrusions (Supplementary Fig. 12a, b, d). The simulation results were consistent with our experiment observations (Supplementary Fig. 12b, c), thus suggesting that hyperdynamic membrane protrusions induced by Rac1 could be stabilized by cell-matrix adhesions as well as adhesion-linked stress fibers. However, all genotypes had no effect on cell migration during the ruffling phase (Supplementary Fig. 12e).

For the simulation effects on the spreading phase, Rho1 inhibition significantly suppressed collective cell P-to-A migration and basal area expansion; differently, Rac1 weak inhibition (escapers often observed in our experiments) caused moderate blockade whereas Rac1 strong inhibition (the one seriously blocking the oocyte growth) theoretically strongly blocked collective spreading and basal area expansion (Fig. 7d, Supplementary Fig. 12f–i and Supplementary Movie 14). This can be explained by the fact that strong inhibition of Rho1 or Rac1 results in the drastic defect in oocyte A–P extension severely hindering passive cell migration, thus emphasizing that oocyte directional growth induced by nurse cell dumping is critical for driving passive collective migration.

**Statistics and Reproducibility**

No statistical method was used to predetermine sample size. No data were excluded from the analyses. The experiments were not randomized. The investigators were not blinded to allocation during experiments and outcome assessment.

All data are presented as mean ± SD. Statistical analysis to compare results between two groups was carried out by the unpaired Student's *t* test, while one-way ANOVA test with Tukey's post-test analysis was used when comparing multiple groups (GraphPad Prism software). A value of $P > 0.05$ was considered to be not significant (ns); a value of $P < 0.05$ (*), $P < 0.01$ (**) or $P < 0.001$ (***) was considered to be differently statistically significant, while a value of $P < 0.0001$ (****) was considered to be remarkably statistically significant.

Exact P values and exact N sample sizes are listed in Supplementary Note 1. The experiments were performed, in general, on the N indicated follicle cells from at least 6 independent egg chambers, or the N indicated oocytes. Optogenetic experiments were performed in the N indicated samples. Choice of all samples is unbiased.

**Reporting summary**

Further information on research design is available in the Nature Portfolio Reporting Summary linked to this article.

## Data availability

Complete data are available in the main article, supplementary materials, and source data files. Since all the raw confocal imaging data supporting the findings of this study runs more than four terabytes and in multiple files, we have not submitted it to the public repository but preserved in our NAS drive and are freely available from the corresponding author (Contact Address: xiaobo.wang@univ-tlse3.fr). Representative images are in the main or Supplementary Figs. Source data are provided with this paper.

## Code availability

The codes used for analyses of follicle cell features and signal intensity (named as: cell feature and intensity) can be found within the website: https://github.com/heishuiguo/cell-and-fiber-feature. The codes used for draw heatmap of curvature on the cell edge (named as: curvature) can be found within the website: https://github.com/heishuiguo/cell-and-fiber-feature. The codes used for analyses of relative angle of stress fibers (named as: polarity) can be found within the website: https://github.com/heishuiguo/cell-and-fiber-feature. The codes used for isolation of D–V oriented fibrous signals (named as: isolation of D–V oriented fibrous signal) can be found within the website: https://github.com/heishuiguo/cell-and-fiber-feature. The custom codes for modeling and simulation are available from the corresponding authors upon reasonable request.

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

## Acknowledgements

The authors thank Adam Martin, Thomas Lecuit, Hong Yang, Yohanns Bellaiche, Bing He, Stefano De Renzis, Jennifer Zanet, Bloomington *Drosophila* stock center and Vienna *Drosophila* RNAi center for flies. We thank Magali Suzanne and Martine Cazales for their technical help in performing laser ablation experiment. We thank Muriel Grammont and Matteo Rauzi for the discussion of manuscript preparation. We thank Shi-Lei Xue for the advice in mathematical modeling. We thank the CBI Toulouse imaging facility at the Université Paul Sabatier. We thank the *Drosophila* facility at CBI Toulouse.  This work is supported by Agence Nationale de la Recherche (ANR PRC AAPG2022 NEW-corset, ANR PRC AAPG2022, ForcesOnCell) to X.W.; National Natural Science Foundation of China (12132004, U19A2006) to Y.L.; National Natural Science Foundation of China (11921002, 12032014) to X.-Q.F.; National Natural Science Foundation of China (12272086) to S.L.; Agence Nationale de la Recherche (ANR-20-CE45-0024, ANR-22-CE42-0026) to T. M.; S.L., S.Z., H.L. and N.S. are supported by China Scholarship Council (CSC) visiting scholar or PhD fellowship; S.Z. is supported by Ligue contre le Cancer PhD fellowship.

## Author contributions

S.L. and X.W. designed the project. S.L., H.L., S.Z. and N.S. performed experiments. V. D. performed the laser ablation experiments. S.L., K.-F.Y. and J.L. developed methods and data analysis scripts. S.L. and J.L. analyzed the data. T.M. performed signal spatial temporal correlation analysis. Z.-Y.L. and X.-Q.F. developed theoretical models and performed simulations. S.L., K.B., X.-Q.F., Y.L. and X.W. wrote the manuscript.

## Competing interests

The authors declare no competing interests.
