## [Peer Review File · Nature Communications]

Basal actomyosin pulses expand epithelium coordinating cell flattening and tissue elongationREVIEWER COMMENTS

Reviewer #1 (Remarks to the Author):

In this manuscript, Li et al use the follicular epithelium of *Drosophila* egg chamber as a model to investigate the role of cortical actomyosin structures in epithelial cell spreading. The authors show that the follicular epithelial cells (FCs) that cover the developing oocyte undergo an expansion wave during their posterior-to-anterior spreading as the oocyte grows and elongates. The expansion wave proceed with three phases associated with specific activities at the basal domain of the FCs: (1) a ruffling phase, which is featured by loss of DV-oriented actomyosin stress fibers and appearance of dynamic membrane ruffles, (2) an area increasing phase, during which the basal cell area rapidly increases as membrane ruffling attenuates, and the pulsatile actomyosin structures start to restore, and (3) a collective spreading phase associated with formation of planar polarized membrane ruffles, restoration of pulsatile actomyosin structures, and a slow increase in basal area. Through quantitative image analyses, the authors further demonstrate close spatiotemporal correlations between basal membrane and cell area dynamics and the dynamics of actomyosin, focal adhesions, and their upstream regulators Rac1 and Rho1. Genetic and optogenetic manipulation of Rac1 and Rho1 or focal adhesions in FCs interferes with specific aspects of basal cell behavior, which in turn influences the AP-oriented elongation of the oocyte. Conversely, inhibiting oocyte growth by blocking nurse cell dumping abolishes the expansion wave of the FCs and strongly inhibited their spreading. Using laser ablation and super-resolution imaging techniques, the authors show that the actomyosin structures during spreading phase are non-contractile and have distinct organization compared to contractile stress fibers. Finally, using a computer vertex model, the authors show that a combined action of FC cell shape changes and polarized oocyte growth can recapitulate the expansion wave as observed during FC spreading.

While the role of actomyosin structures in generating contractile forces during cell and tissue morphogenesis has been well established, it is less clear how actomyosin structures may facilitate cell area expansion. By studying the dynamics and regulation of actomyosin during epithelial spreading, the authors provide important new insights into this question. The experiments are well designed, and the data acquisition and quantitative analyses are elegant and of high quality. My main questions lie in the interpretation of some of the data, in particular the proposed expansion function of actomyosin and the role of FC cell shape change in driving FC spreading. My detailed comments are listed below.

Major comments:

1. One of the main conclusions of the manuscript is that actomyosin can display “expanding properties, in addition to conventional contractile properties”. Whereas the way actomyosin generates contractile forces has been well illustrated in many systems, the mechanism by which actomyosin supports FC expansion and spreading is not clear. The authors show that during the expansion wave, local actomyosin activation typically follows cell area expansion rather than preceding it, in contrast to the scenario when actomyosin drives cell constriction. These observations suggest that during FC spreading, actomyosin structures function to stabilize or reinforce basal expansion, rather than generating the “expansion forces” that physically enlarge the basal domain. This mode of action is distinct from actomyosin-mediated contractile behavior where actomyosin provides tensile forces that directly drive cell constriction. It is therefore a little misleading to draw close parallels and contrast between the contractile function of actomyosin and the function observed in this study. The

term “expanding properties” should be more clearly defined to avoid confusion.

2. The authors show that when nurse cell dumping is blocked, FC spreading is nearly completely abolished (Fig. 5l,m). This result raises the question of to what extent the basal cell behaviors mediated by Rac1 and Rho1 can drive collective cell spreading in the absence of oocyte growth. The authors’ data seem to support a scenario where nurse cell dumping and the resulting oocyte volume increase provide the main driving forces that result in “passive” FC spreading, whereas the changes in the basal cell properties in the follicular epithelium (mediated by Rac1 and Rho1) mainly function to reduce the resistance as the oocyte elongates, and/or stabilize cell shape changes in a planar polarized fashion to facilitate AP-oriented oocyte elongation. A question related to this point is whether the activation of Rac1 and Rho1 in FCs is dependent on oocyte elongation, which could be tested by examining the extent of Rac1 and Rho1 activation in the absence of nurse cell dumping. The result of this experiment might help to further elucidate the relative contribution of the “active” (Rac1 and Rho1 mediated) and the “passive” (oocyte elongation mediated) mechanisms in FC spreading.

3. Among the three phases of the expansion wave, the increasing phase features the highest rate of basal area expansion and cell height reduction. However, the spreading phase has the highest rate of AP-directed FC spreading (Fig. 1j). It is puzzling why the highest rate of basal area expansion and the highest rate of cell spreading do not occur in the same phase.

4. The physical characteristic of the expansion wave should be described in more detail. For example, for cells at different locations of the wave region, what is the typical time when they initiate each of the three phases of the expansion wave? Showing this dynamic information (e.g., by heatmaps) will help demonstrate the rate and direction of wave propagation more clearly.

5. When Rac1 function is disrupted, the FCs show defects in both the ruffling phase and the spreading phase. The authors propose that Rac1 has distinct functions in these two phases that promote membrane ruffling and directional membrane expanding, respectively. However, it is unclear whether the defect in the spreading phase is solely due to the earlier defect in the ruffling phase. The authors should address this question, either by performing stage specific inhibition of Rac1 using optogenetic Rac1DN or discussing different possibilities in the text.

6. Given that the rates of basal area expansion and cell height reduction peak during the increasing phase, the dynamics of actomyosin and focal adhesion during this phase should be demonstrated (including images similar to Fig2j and 2m).

7. The authors propose that during the spreading phase “the Rac1 and Rho1 activities in follicle cell basal domains appear to form an integrated signalling system to coordinate membrane ruffles, localized stress fibers and their associated focal adhesions, thereby promoting cell expansion behaviour”. Focal adhesions are well-known to function as “anchors” during cell migration and cell spreading. The authors should discuss the similarities and differences between the function of focal adhesions during FC spreading and the canonical function of focal adhesions during cell spreading.

8. In the example shown in Fig. 2h (“spreading phase”), it appears that the peaks of active Rho1 and F-actin correspond to a temporary pause of cell area increase (8 – 10 min). Is this

a typical observation? If so, does it indicate that pulses of Rho1 activation and F-actin assembly temporally restrict cell area increase during the spreading phase?

9. What is the nature of the F-actin and myosin flow shown in Fig. 6f and Fig. S8? Is it due to movement of existing actomyosin structures, or due to turnover of the actomyosin filaments (formation of new filaments and disassembly of old filaments)? Do focal adhesions flow in a similar manner as F-actin and myosin?

10. The rationale for recapitulating the Rho1DN (and Rac1DN) phenotype using combined factor simulation is not entirely clear. The combined factor simulation seems to assume that Rac1DN or Rho1DN expression in the FCs not only influences FC shape changes but also independently impacts the A-P extension of the oocyte. Such a function is not intuitive since Rac1DN and Rho1DN were expressed in the FCs but not in the oocyte or nurse cells. Some potential explanation should be provided.

Minor points:

1. Both Rho1DN and talin RNAi show defects in the formation of ruffles (Fig. S6c-e). However, neither of the conditions show defects in cell circularity during the ruffling phase. What might be the cause of this apparent discrepancy?
2. Supplemental movie 1: the Myo II signal in the middle region of the tissue appears dimmer than the signal at the peripheral region. Is this due to region-specific regulation of myosin activation or due to technical reasons (e.g., imaging of a single focal plane within the curved egg chamber)? How might this signal heterogeneity influence the quantification of the Myo II signal?
3. How membrane ruffle is defined and quantified (e.g., Fig S6d,e) should be described in the Methods. It is not obvious to me that during the spreading phase the generation of membrane ruffles is biased in the A-P axis. Some annotation or quantification would be helpful.
4. Oocyte growth is faster in PaxOE compared to the control, however, the final oocyte length_{AP}/length_{DV} is comparable between PaxOE and the control (Fig. 4j,k). Why is this the case?
5. How the role of nurse cell dumping was implemented in the simulation should be described in more detail in the Methods.
6. Line797: The MATLAB-based method for correcting photobleaching should be described and justified.
7. "Rac1DN Strong" and "Rac1DN Weak" are only described in combined factor simulations but not in single factor simulations. Some explanation would be helpful.
8. The dashed lines in Fig. 2b,e,h and Fig. 5e are not defined.
9. Fig. 4b: the four phases should be presented following the temporal order to keep consistency with other figures.

Reviewer #2 (Remarks to the Author):

The paper by Li et al. examines the cytoskeletal mechanisms of follicle cell flattening that accompany *Drosophila* oocyte growth. The authors begin by characterizing the phases of follicle cell flattening, which proceeds as a sequence of different cell behaviors, including: membrane ruffling, myosin increasing, and cell spreading phases. The authors, then proceeded to carefully characterize the activity levels and dynamics of Rac1, Rho1, actomyosin, and Talin at these different phases and to tested their roles at different stages by expressing dominant negative Rac1 and Rho1 and Talin/Paxillin perturbations. The authors, then tested the relationship between follicle cell area expansion and oocyte growth. They found that follicle cell flattening was required for proper oocyte expansion and that oocyte expansion was also important for driving follicle cell flattening. Finally the authors characterized the cytoskeletal organization transitions between these phases and developed a theoretical model of epithelial expansion and its coordination with oocyte growth that reinforced the model that both active and passive migration are important to explain the authors results. Overall, I felt that most of the authors conclusions were supported by their data and they did an excellent job at thoroughly quantifying their observations. To my knowledge this is the most detailed description of this cell flattening process. My most significant comments are related to clarity of presentation as outlined in my point-by-point response below.

Main comments:

- There are multiple potential causes of follicle cell flattening in this system, as presented in different places in the text: outward pressure from the expanding oocyte, adhesion between follicle cells and the lengthening oocyte, and potential forces from the actomyosin networks themselves. It's sometimes unclear how separable these sources are, which makes it difficult to evaluate the claim that actomyosin networks drive cell expansion in the system. This is something that I think is worth addressing in the Discussion.
- The simulation results in which Rho1 inhibition only inhibits P-A migration when the oocyte directional growth reduction is included are interesting but raise questions regarding the role of Rho1 in cell spreading. Do the authors have an argument for why Rho1 reduction alone is not sufficient to reproduce the experimental results? Does Rho1 inhibition in the follicle cells somehow inhibit nurse cell dumping, which then inhibits follicle cell expansion?
- In general, the text did not guide the reader through each figure as well as it could have. For Figure 1, the figure callouts to 6-9 figure panels at a time were confusing because the authors didn't explain what each measurement was and why it was important. For Figure 5, there were similar massive callouts without explanation Lines 252-255. I think the paper would be more easily consumable if the authors provided more guidance for their analysis.

Minor comments

- Fig. 1c: colors for Increasing and Ruffling are reversed relative to the rest of the figure panels.
- 'our results support Rac1 and Rho1 signalling as key factors controlling the ruffling and spreading process, respectively, to enlarge cell basal surface area': it seems Rac1 is also important for spreading (Fig. 3i), It seems that the authors should not conclude so strongly that the two GTPases have such cleanly separate activities here.
- Lines 176-178, The defect in basal area could also reflect a defect in migration.
- Lines 183-186: Need the word "respectively" added. E.g. "promoted or inhibited collective

cell spreading process, respectively, . . .”

- Line 207-8: This sentence was confusing, particularly the term ‘focal adhesion continuum’
- Lines 199-200: How does ‘enhanced focal adhesions’ translate into the stability of focal adhesions? The authors show that there can be two different focal adhesion states – one static and one more dynamic.
- Fig. 4b: are Increasing and Spreading data reversed in the plot or in the legend?
- Lines 230-233: Can the authors explain why Rac1 inhibition causes significantly delayed cell flattening while only moderately repressing oocyte A-P directed growth?
- Fig. 3c caption: would “Paxillin-overexpressing” instead of “paxillin-expressing” be more accurate?
- Fig. 4j: presenting the outlines with different maximum elapsed time is slightly confusing. Could the authors either extend the times used for Pax OE and Talin RNAi to 100 min or explain why those times cut off at 60 min?
- Fig. 6a, b: Is myosin intensity equivalent or different between the phases?
- Line 291: Sentence requires “respectively” E.g. “pre-wave or spreading phase, respectively, consistent with . . .”
- Line 295-296: Just 1 sentence should not be a paragraph.
- Acknowledgements: “Jennifer Zallet” should be “Jennifer Zallen”

-Adam Martin

I sign my review to promote transparency and accountability in the review process.

Reviewer #3 (Remarks to the Author):

The authors have examined how follicle cells manage to expand and flatten starting from a state where they form a corset around the oocyte characterized by strong pulsatile actomyosin networks that could be expected to preclude such an expansion. They show that, concomitant with nurse cell dumping and oocyte growth, follicle cells first develop a strong Rac1 activity basally (measured with a PAK3RBD fluorescent reporter) which promotes cell ruffling, then a RhoA activity (measured with AniRBD fluorescent reporter) leading to enhanced focal adhesion formation. They further show that Rac1, RhoA and focal adhesion are required for follicle cell spreading, and that dumping is also essential. Finally, they examine the relative organization of actin and myosin II and offer a mathematical model accounting for their results.

Taken together, their results provide a rather detailed mechanistic description of the cytoskeletal reorganization accounting for how follicle cells adapt to the oocyte expansion by flattening and migrating. Their data nicely describe the transition between Rho and Rac signaling. The images and statistics are convincing. More generally, this study is important because it falls in the general framework of how two tissues can interact mechanically during morphogenesis, and opens many new questions for the understanding of oocyte-follicle mutual interactions. I strongly support publication of this work after addressing three issues.

1/ The temporal model of Fig. 2p, along with the text, suggests that Rac signaling is the first key event. However, unless mistaken, Rho signaling is strong until stage 10 to stimulate the basal Myo2 pulses. So, I would have imagined that the first key event once dumping starts should be Rho down-regulation, which the authors did not consider. If however it is Rac1 that induces RhoA downregulation, how does this occur? It would be useful to cross-examine RhoGAPs and RacGEFs and their activity at the stage 10 transition. The Bellaïche lab very recently published a systematic study of all Rho/Rac GAPs and GEFs present in

follicle cells which should be useful.

2/ Related to the issue of the respective and mutual mechanical interactions between the oocyte and follicle cells, the authors should briefly speculate on the signaling process leading to Rac1 and RhoA activity up- or down-regulation, as well as to the signaling preventing oocyte expansion when follicle ruffling or spreading is hampered. At the mesoscopic level, but not at the signaling and cellular levels, it is reminiscent of the situation encountered during *C. elegans* late elongation which depends on a mechanical cross-talk between muscles and the epidermis.

3/ The presentation of the final model should be improved. For readers who would not want to go through the Methods section, it is not easy to understand what was done just reading the main text and Fig. 7 legend. Please indicate

- what are the red lines in 7c & their respective intensity in 7d (how do they relate to stress fibers);
- what are the graded grey rectangles in 7e, which are unchanged in all conditions except in dumping
- describe also briefly the speedometer
- it is not clear what has been combined in Fig. S11
- the comparison between the single-factor simulations and the actual experimental data do not entirely match for most parameters either in absolute terms (as they point out for Rho1DN) or in relative terms; this should be better commented. It might for instance indicate that Rac1 influences RhoA and vice-versa.

Minor:

- Author contributions: the contributions of Jiaying Liu (JL), Kaifu Yang (KY), Vanessa Dougados (VD) and Thomas Mangeat (TM) have not been listed in the relevant section.
- Line 348-354: the words adaptation and adaptation were used indifferently; it would be better to stick to a single word.
- Fig. 1 legend: the duration of each phase as indicated in the main text (lines 94-100) or in panel bright are not the same; please clarify.
- Fig. 2bc legend: are fold changes measured over the entire area of the cell or over a small subsection? Please clarify here and in the methods section.
- Fig. 2pq legend: what is the time unit in those panels?
- Fig. 2e,h: the respective peaks of actin versus Rac1 or Rho1 differ by about 1 min, which is much higher than those reported in the germband or in early *C. elegans* embryos (in the range of a few seconds). Could you comment.
- Fig. 1c versus Fig. 2b,e,h,k,p: it is not easy to juxtapose the myosin-2 fluctuations from Fig. 1c with those of actin in Fig. 2, despite the fact that Fig. S1 shows that actin and myosin-2 are fully cross-correlated.
- Fig. 4b: define the "flatten ratio" (line 470)

To make it easier for reviewers to track our changes, we mark new insertions in yellow and deletions as strikethroughs.

Reviewer #1 (Remarks to the Author):

Major comments:

1. One of the main conclusions of the manuscript is that actomyosin can display “expanding properties, in addition to conventional contractile properties”. Whereas the way actomyosin generates contractile forces has been well illustrated in many systems, the mechanism by which actomyosin supports FC expansion and spreading is not clear. The authors show that during the expansion wave, local actomyosin activation typically follows cell area expansion rather than preceding it, in contrast to the scenario when actomyosin drives cell constriction. These observations suggest that during FC spreading, actomyosin structures function to stabilize or reinforce basal expansion, rather than generating the “expansion forces” that physically enlarge the basal domain. This mode of action is distinct from actomyosin-mediated contractile behavior where actomyosin provides tensile forces that directly drive cell constriction. It is therefore a little misleading to draw close parallels and contrast between the contractile function of actomyosin and the function observed in this study. The term “expanding properties” should be more clearly defined to avoid confusion.

Answer: We agree with the reviewer. Now we define the term “expanding properties” in the abstract (page 2, yellow label), as follows:

“Mechanics, structures and biological function implicate that actomyosin networks function to stabilize or reinforce basal expansion, rather than to generate expansion forces, in order to exert their expanding properties.”

We also include some discussion in the Discussion section (page 17, yellow label), as follows:

“Notably, these novel expanding properties of basal stress fibers do not appear to generate expanding forces that physically enlarge follicle cell basal domain; instead, activation of stress fiber actomyosin networks typically occurs following basal cell area expansion, supporting their roles in stabilizing or reinforcing basal expansion.”

2. The authors show that when nurse cell dumping is blocked, FC spreading is nearly completely abolished (Fig. 5l, m). This result raises the question of to what extent the basal cell behaviors mediated by Rac1 and Rho1 can drive collective cell spreading in the absence of oocyte growth. The authors’ data seem to support a scenario where nurse cell dumping and the resulting oocyte volume increase provide the main driving forces that result in “passive” FC spreading, whereas the changes in the basal cell properties in the follicular epithelium (mediated by Rac1 and Rho1) mainly function to reduce the resistance as the oocyte elongates, and/or stabilize cell shape changes in a planar polarized fashion to facilitate AP-oriented oocyte elongation. A question related to this point is whether the activation of Rac1 and Rho1 in FCs is dependent on oocyte elongation, which could be tested by examining the extent of Rac1 and Rho1 activation in the absence of nurse cell dumping. The result of this experiment might help to further elucidate the relative contribution of the “active” (Rac1 and Rho1 mediated) and the “passive” (oocyte elongation mediated) mechanisms in FC spreading.

Answer: We agree with the reviewer. Under physiological conditions, dumping and follicle cell spreading occur simultaneously or interact with each other. On the one hand, the increase in basal area caused by dumping provides space preparation for the spreading of follicle cells. On the other

hand, dumping-enhanced oocyte pressure (from our unpublished data) might provide stimulatory factors to regulate Rac1 and Rho1 signaling. Thus, we postulate that theoretically, dumping is the prerequisite for spreading, and thus we consider dumping to be a “passive” driver. That is also shown in our mathematical model.

The question “whether the activation of Rac1 and Rho1 in FCs is dependent on oocyte elongation” is beneficial to elucidate the relative contribution of the “active” and the “passive” mechanisms in follicle cell spreading. Therefore, we analyzed the activation and distribution of Rac1 and Rho1 following dumping inhibition. See our Rebuttal Fig. 1 and our following conclusion:

When dumping was inhibited, neither the activity nor the distribution of Rac1 was significantly altered in individual follicle cells, compared with the follicle cells in control tissues (Rebuttal Fig. 1a, b); Rac1 activity in dumping-deficient tissues also increased during both the ruffling and spreading phases (compared to the pre-wave phase), resulting in the randomly oriented ruffles or P-A oriented protrusions, all of which phenocopied follicle cells in control tissues (Rebuttal Fig. 1a, b). However, Rac1 activity increase occurred in all follicle cells of dumping defective tissues and was not restricted to the posterior regions observed in control tissues (we observed Rac1 activity increase in most of follicle cells in *Singed* mutant tissues).

Different from subcellular Rac1 activity, Rho1 activity normally located in medial basal regions has moved to the cell periphery/junctions in follicle cells during the spreading stage, when dumping was inhibited (Rebuttal Fig. 1c, d), while no prominent changes were observed during pre-wave and ruffling phases (data not shown). Translocation of Rho1 activity thus might account for the loss of DV stress fibers and actomyosin networks and thus the defects in follicular cell spreading.

Although we found these interesting observations of basal Rac1 activity (spreading Rac1 activation to most follicle cells) and basal Rho1 activity (translocation from medial basal regions to junctions), we have limited understanding about how dumping modulates Rac1 and Rho1 basal activities in follicle cells. Indeed, it is difficult for us to answer the question (the relative contribution of the “active” and the “passive” mechanisms) only through biological inhibition strategies, because “active” and “passive” mechanisms are tightly coupled. Based on current data and phenomena, in our opinion, “passive” signals might be the trigger of “active” behaviours and at the same time are also regulated by the feedback of “active” signals; both “active” and “passive” mechanisms are indispensable for promoting follicle cell spreading.

Considering this complexity between “active” and “passive” signals, we didn’t include Rebuttal Fig. 1 (the effect of dumping on Rac1/Rho1 activity) in our current revision. If the reviewer feels it important to our manuscript, we will update it into a supplementary figure.

Rebuttal Figure 1. Basal Rac1 and Rho1 activities in dumping-deficient tissues compared to control tissues.

a) Representative images of PAK3-RBDGFP (Rac1 activity reporter) and LifeAct-RFP (F-actin reporter) in follicle cells of WT tissues vs. Singed mutant tissues (dumping-deficient tissues), during 3 indicated phases. Heatmap is used to better view relative level of PAK3-RBDGFP in follicle cells. RBD colour marks the relative level of PAK3-RBDGFP. **b)** Quantification of relative Rac1 activity in follicle cells of WT tissues vs. Singed mutant tissues during different phases. **c)** Representative images of Ani-RBDGFP (Rho1 activity reporter) and LifeAct-RFP (F-actin reporter) in follicle cells of WT tissues vs. Singed mutant tissues during spreading phase. Heatmap is used to better view relative level of Ani-RBDGFP in follicle cells. RBD colour marks the relative level of Ani-RBDGFP, and white dotted lines mark basal junctions (clearly showing translocation of Rho1 basal activity to basal junctions in Singed mutant tissues). **d)** Quantification of Rho1 activity ratio between medial basal regions and basal junctions in follicle cells of WT tissues vs. Singed mutant tissues during spreading phase. Scale bars are 10 microns in (a) and (c).

3. Among the three phases of the expansion wave, the increasing phase features the highest rate of basal area expansion and cell height reduction. However, the spreading phase has the highest rate of AP-directed FC spreading (Fig. 1j). It is puzzling why the highest rate of basal area expansion and the highest rate of cell spreading do not occur in the same phase.

Answer: We propose that high basal area expansion rate represents the rapid cell morphological deformation and flattening of follicle cells, while high cell spreading rate means the relatively fast movement of follicle cells on the matrix. During the increasing phase, after the pre-accumulation of cell membranes during the ruffling phase, follicle cells undergo rapid morphological deformation, with the basal area rapidly expanding and the height decreasing sharply; and this cell deformation and flattening process occur over a short period of time, even without prominent directional movement. However, during the spreading phase, follicle cells undergo rapid AP-directed migration as the oocyte elongates, and this further flattening and expansion process occurs over a much longer time period (with a smaller expansion/flattening rate). This difference thus explains why the highest rate of basal area expansion and the highest rate of cell spreading do not occur in the same phase.

4. The physical characteristic of the expansion wave should be described in more detail. For example, for cells at different locations of the wave region, what is the typical time when they initiate each of

the three phases of the expansion wave? Showing this dynamic information (e.g., by heatmaps) will help demonstrate the rate and direction of wave propagation more clearly.

Answer: We agree with the reviewer. To describe the physical characteristics of epithelial cell expansion waves in more detail, a cartoon of follicle cells at different locations in the egg chamber is now included in Supplementary Fig. 2j, as shown below.

5. When Rac1 function is disrupted, the FCs show defects in both the ruffling phase and the spreading phase. The authors propose that Rac1 has distinct functions in these two phases that promote membrane ruffling and directional membrane expanding, respectively. However, it is unclear whether the defect in the spreading phase is solely due to the earlier defect in the ruffling phase. The authors should address this question, either by performing stage specific inhibition of Rac1 using optogenetic Rac1DN or discussing different possibilities in the text.

Answer: We thank the reviewer for pointing out this interesting question. As the reviewer suggested, we include a discussion of different possibilities in the Discussion section (page 19-20, yellow label), as follows:

“Another unclear point is the unique function of Rac1 in controlling ruffling and spreading processes. Strong basal Rac1 activities, observed during the ruffling and spreading phases, are distributed at random ruffles and A-P oriented protrusions, respectively, thereby implicating different roles of Rac1 during these 2 phases. Although Rac1 is critical during the ruffling phase to drive the dynamics of random ruffles, there is no conclusive evidence that ruffling phase defect driven by Rac1 inhibition can directly inhibit the spreading phase. Some evidence indicates that ruffling behaviour of follicle cells can help the cell spreading process simple and orderly. For example, experiencing ruffling makes follicle cells more susceptible to metamorphosis and remodelling of the cytoskeleton and focal adhesions, and also prepares the cell with sufficient membranes to allow rapid expansion. On the other hand, during cell spreading process, we observed that the first step in the pulsatile motion of follicle cells is the formation of protrusions at the anterior edge; and signal correlation supports that these leading protrusions should be driven by Rac1. Logically, we regard the ruffling phase as a bottle-neck in egg chamber development, and only follicle cells that successfully pass the ruffling can enter the subsequent process normally, as we observed in the phenotype of Rac1 strong inhibition. A possible explanation is that follicle cells cannot decouple from the ECM without going through the ruffling phase; as the tension associated with egg chamber growth increases, the original DV-directed stress fibers during pre-wave phase will be destroyed, and the structure of epithelial monolayer and egg chamber may be compromised.”

6. Given that the rates of basal area expansion and cell height reduction peak during the increasing phase, the dynamics of actomyosin and focal adhesion during this phase should be demonstrated

(including images similar to Fig2j and 2m).

Answer: In the updated Supplementary Fig. 4r-t, we include this important information, as shown below. This update is also included in the text (page 8, yellow label), as follows:

“during the ruffling, increasing and spreading phases, focal adhesion densification also exhibited a pulsatile pattern following basal actomyosin networks (Fig. 2j-o and Supplementary Fig. 4r-t).

7. The authors propose that during the spreading phase “the Rac1 and Rho1 activities in follicle cell basal domains appear to form an integrated signalling system to coordinate membrane ruffles, localized stress fibers and their associated focal adhesions, thereby promoting cell expansion behaviour”. Focal adhesions are well-known to function as “anchors” during cell migration and cell spreading. The authors should discuss the similarities and differences between the function of focal adhesions during FC spreading and the canonical function of focal adhesions during cell spreading.

Answer: We thank the reviewer for pointing out this interesting comparison. As the reviewer suggested, we include a discussion of the similarities and differences between the structure and function of focal adhesion during follicle cell spreading and canonical focal adhesion during mesenchymal cell spreading in the text (page 17-18, yellow label); we also present Rebuttal Fig. 2 (Supplementary Fig. 8c in the text, also see in our response to comment 9) to help visualize focal adhesion dynamics during follicle cell spreading; both as follows:

“It is well known that focal adhesions function as “anchors” during cell migration and spreading²¹. Here, our observed expanding stress fibers demonstrate some similarities and differences with the typical structures and functions of focal adhesions during cell spreading. According to different subcellular localization and termination sites, actin stress fibers are divided into dorsal stress fibers, transverse arc and ventral stress fibers²¹. Dorsal/ventral stress fibers are anchored to the matrix surface via focal adhesions²¹. Establishing the orientation of the filaments in stress fibers is critical because it defines the contractile properties and structures, as well as focal adhesion distributions and functions⁴⁰. In 2D cultured mesenchymal migrating cells, dorsal/ventral stress fibers show a polar distribution along the direction of cell movement; correspondingly, focal adhesions linked with dorsal or ventral stress fibers are distributed at the cell front or rear, respectively⁴¹. Differently, in follicle cells on the curved surface during the spreading phase, the predominant form of actin stress fibers is the ventral stress fiber perpendicular to the A-P direction of migration; correspondingly, focal adhesions are distributed on both sides of the D-V axis near the edge. Due to the different orientations of stress fibers, the direction of mature focal adhesions of mesenchymal migrating cells is elongated along the direction of cell migration^{42,43}, while the direction of mature focal adhesions in follicle cells is perpendicular to the A-P migration direction. Thus, the directional difference in focal adhesions and actomyosin networks leads to an adaptation of cell motility strategies. The turnover of dispersed focal adhesions in mesenchymal migrating cells requires the coordinated dynamics of 3 different stress fibers to sequentially complete the processes of focal

adhesion assembly, maturation and disassembly⁴⁰. However, in spreading follicle cells, DV-oriented actomyosin and focal adhesions assemble and disassemble at the anterior or posterior terminal of the cell, respectively, allowing cell movement in the P-to-A direction. To perform directional migration, these two types of cells use their respective strategies to establish the traction difference between the front and rear of the cell, and to establish continuous, coordinated formation and turnover of these focal adhesions, thereby anchoring or releasing cell adhesions at the cell front or rear.”

Rebuttal Figure 2. Dynamics of focal adhesions and basal stress fibers during follicle cell spreading. Representative time-lapse images of Talin-GFP and LifeAct-RFP showing how the DV-oriented actomyosin and focal adhesions assemble and disassemble at the anterior or posterior terminal of follicle cells, respectively, allowing cell movement in the P-to-A direction. Asterisk marks new assembly of focal adhesions and stress fibers at cell anterior edge, and marks gradual disassembly of focal adhesions at cell posterior edge. To analyse the signals dynamic at the same area within the cell, we corrected the position of cells in the image so that the cells did not show relative displacement in the images at different time points. Scale bars are 10 microns in all figures.

8. In the example shown in Fig. 2h (“spreading phase”), it appears that the peaks of active Rho1 and F-actin correspond to a temporary pause of cell area increase (8 – 10 min). Is this a typical observation? If so, does it indicate that pulses of Rho1 activation and F-actin assembly temporally restrict cell area increase during the spreading phase?

Answer: We apologized for some confusing information here. Here, the first maximum peak of F-actin signals follows the previously pulsatile Rho1 signals that was missed in Fig. 2h (before and within 4 minutes of the curve); and the second maximum peak of F-actin signals follows the pulsatile Rho1 signals that occurs between 4 and 14 minutes of the curve. The three-color dotted line shows the peaks we are concerned about, as shown below.

We thank the reviewer for pointing out this interesting question. The phenomenon is typical of observations, but we do not think that pulses of Rho1 activation and F-actin assembly temporally restrict cell area increase during the spreading phase. On the contrary, Rho1 activation and F-actin assembly seem to stabilize cell structure by helping the formation of focal adhesions in the newly spreading area. In the absence of pulsed Rho1 activation and F-actin assembly, basal cell area retracts back to the state before this cell expansion.

9. What is the nature of the F-actin and myosin flow shown in Fig. 6f and Fig. S8? Is it due to movement of existing actomyosin structures, or due to turnover of the actomyosin filaments (formation of new filaments and disassembly of old filaments)? Do focal adhesions flow in a similar manner as F-actin and myosin?

Answer: Based on the dynamics of actomyosin signals, we propose:

“Interestingly, our analysis of signal dynamics supports that the nature of basal actomyosin network dynamics is due to actomyosin filament turnover, i.e., the formation of new filaments at the cell front and the disassembly of old filaments at the cell rear; and focal adhesions also show a similar pattern to basal actomyosin networks (Supplementary Fig. 8c).” Please see our update text (page 14, yellow label) as well as Supplementary Fig. 8c.

10. The rationale for recapitulating the Rho1DN (and Rac1DN) phenotype using combined factor simulation is not entirely clear. The combined factor simulation seems to assume that Rac1DN or Rho1DN expression in the FCs not only influences FC shape changes but also independently impacts the A-P extension of the oocyte. Such a function is not intuitive since Rac1DN and Rho1DN were expressed in the FCs but not in the oocyte or nurse cells. Some potential explanation should be provided.

Answer: In simulation, oocyte A-P extension and the strength of Rho1/Rac1 are independent parameters which are manipulated separately. Therefore, the dynamics of follicle cells including ruffling, increasing and spreading are comprehensive effects of all factors. In simulation, we could manipulate those factors individually (single-factor simulation) or systemically (combined-factor simulation), whereas in experiments we observed that oocyte A-P extension is significantly affected in Rac1DN/Rho1DN condition. Thus, although there is currently no conclusive evidence as to whether Rac1DN/Rho1DN has a direct effect on nurse cell dumping or oocyte A-P extension, to investigate the effects of dumping-induced oocyte A-P extension on follicle cell dynamics, we implanted the experimental data observed in Rac1DN/Rho1DN phenotype in the combined-factor simulation and underlined the importance of passive migration. As in biology why Rac1DN/Rho1DN induce the defects in oocyte A-P extension, we speculate that dumping process is regulated by some feedback from Rac1DN- or Rho1DN- mediated inhibition of follicle cell spreading. Slowed or suppressed dumping process might undergo some adjustments, such that oocyte A-P extension is inhibited while D-V extension is increased, which is also supported by our results: A-P length decreased but D-V length increased in Rac1DN and Rho1DN egg chambers (Figure 4j). To fully answer this question, more experiments are needed to decouple the behaviors of follicle cells from nurse cells, which was not the focus of this study.

Minor points:

1. Both Rho1DN and talin RNAi show defects in the formation of ruffles (Fig. S6c-e). However, neither of the conditions show defects in cell circularity during the ruffling phase. What might be the cause of this apparent discrepancy?

Answer: We propose that dynamic ruffle formation is regulated by Rac1 signalling, which can be easily reflected by low circularity parameter. However, this cell circularity status cannot reflect the ruffle stability, and our studies reveal that Rho1 and focal adhesions are primarily involved in the maintenance of ruffle stability.

In Fig. S6c, cell outlines showed that PaxOE, Rho1DN and Talin RNAi do not have much effect on ruffle formation, whereas asterisk-marked ruffles demonstrated that they can strongly alter ruffle size and stabilization time (15 minutes in PaxOE, but less than 5 minutes in Rho1DN/Talin RNAi, also quantified in Fig. S6d-e).

To make this difference clear, we now include all these explanations in the text (page 10, yellow label), and also as follows:

“Here, dynamic ruffle formation, reflected as the circularity index, is mainly regulated by Rac1 signalling, whereas Rho1 and focal adhesions are primarily involved in the maintenance of ruffle timing and stability.”

2. Supplemental movie 1: the Myo II signal in the middle region of the tissue appears dimmer than the signal at the peripheral region. Is this due to region-specific regulation of myosin activation or due to technical reasons (e.g., imaging of a single focal plane within the curved egg chamber)? How might this signal heterogeneity influence the quantification of the Myo II signal?

Answer: Supplementary Movie 1 showed a dorsal view of the egg chamber where cells in the middle region of the tissue do not undergo the wave process (which we discussed in Fig. 11). We did not use these cells (in the middle region of dorsal view) to study wave behavior, and thus this signal heterogeneity did not affect our analysis of the data in this study.

To clarify this region specificity, we now update this information in supplementary information (page 16, yellow label), and also as follows: “Dorsal view time-lapse sequence of ...”

3. How membrane ruffle is defined and quantified (e.g., Fig S6d, e) should be described in the Methods. It is not obvious to me that during the spreading phase the generation of membrane ruffles is biased in the A-P axis. Some annotation or quantification would be helpful.

Answer: We now include the definition and quantification of membrane ruffles in the Method section (the curvature of the cell edge, page 41; the orientation of ruffle, page 42; the circularity value, page 42), suggested as the reviewer.

We also include the quantification of the ruffle directions in Fig. 1g, as shown below.

4. Oocyte growth is faster in PaxOE compared to the control, however, the final oocyte length_{AP}/length_{DV} is comparable between PaxOE and the control (Fig. 4j, k). Why is this the

case?

Answer: We observed that Pax-OE tissues have a faster oocyte growth rate than the control tissues, so that Pax-OE tissues reach the maximal elongation status around 60 minutes (and then cannot elongate further), compared to the control tissues (over 80 minutes in their oocyte elongation process).

5. How the role of nurse cell dumping was implemented in the simulation should be described in more detail in the Methods.

Answer: We now include the details of how nurse cell dumping was implemented in the simulation in the Method section (page 48, yellow label), suggested as the reviewer. The detailed description is as follows:

“During late stages of *Drosophila* oogenesis, nurse cells rapidly transfer their cytoplasmic contents into the oocyte, thereby increasing the oocyte length along the A-P axis. As the oocyte grows along the A-P axis, follicle cells deform and migrate on the oocyte surface. In the simulation, we introduced the effects of nurse cell dumping by modulating the A-P growth rate of the oocyte.”

6. Line797: The MATLAB-based method for correcting photobleaching should be described and justified.

Answer: We thank the reviewer for pointing out this missing information. Now we include the detailed information of correcting photobleaching in the Method section (page 40-41, yellow label), as follows:

“Normally, in the absence of significant photobleaching (less than 5% decrease of intensity between two adjacent frames, especially for background signals), we did not perform a correction, but it must be ensured that under these conditions a rising signal (such as myosin-II, Rac1, Rho1 and focal adhesions) can be observed during these four phases. Once an excessive difference of intensity was detected, after confirming that it was not due to a phase conversion, we performed a correction based on two adjacent frames (especially for background signals). Although this correction is not perfect, it is sufficient to allow us to observe upward and downward fluctuations of most signals even affected by photobleaching.”

7. “Rac1DN Strong” and “Rac1DN Weak” are only described in combined factor simulations but not in single factor simulations. Some explanation would be helpful.

Answer: Due to the limitations of genetic manipulation (genetic escapers), we experimentally found that Rac1DN produces varying degrees of repression, specifically, with strong and weak phenotypes. Strong inhibition of Rac1 prevents follicle cells from entering the following wave process and stops oocyte growth. Therefore, to investigate the regulatory effect of Rac1 on the follicle cell wave and migration, we focused on the Rac1DN weak phenotype and referred to the Rac1DN weak phenotype as Rac1DN in analyzing the experimental data. In single-factor simulation, we want to individually investigate the effects of Rac1DN on both ruffling and spreading, so we choose Rac1DN weak phenotype to simulate (to avoid our previous confusion, we update Rac1DN to Rac1DN weak in our current revision, page 9, yellow label “Note that...we focused on Rac1DN weak phenotype and refer to the Rac1DN weak phenotype simply as Rac1DN in the following context.”). However, since Rac1DN strong phenotype almost stops the oocyte growth, combined-factor simulation is necessary to reproduce the experimental observations. We have added a detailed description of the difference

between Rac1DN weak and strong phenotype in the manuscript.

8. The dashed lines in Fig. 2b, e, h and Fig. 5e are not defined.

Answer: We now include the definition of the dashed lines in Fig. 2b, e, h and Fig. 5e, suggest as the reviewer. This definition is listed as follows:

In Fig. 2, “The dashed lines in (b, e, h, k, n) indicate the peaks of corresponding colour-marked signals within one cycle.”

In Fig. 5, “The dashed lines in (e, g, i) indicate the starting time points of ruffling process.”

In addition, we also include the definition of the dashed lines in Fig.2 (k and n), Fig. 5 (f-j), Sup Fig.1c, Sup Fig. 2(d, f, g, n and o), Sup Fig.3 (c, d, f, g and o), Sup Fig.5c, Sup Fig.9a and Sup Fig.11c.

9. Fig. 4b: the four phases should be presented following the temporal order to keep consistency with other figures.

Answer: In the updated Fig. 4b, we correct this as suggested by the reviewer.

To make it easier for reviewers to track our changes, we mark new insertions in yellow and deletions as strikethroughs.

Reviewer #2 (Remarks to the Author):

Main comments:

1. There are multiple potential causes of follicle cell flattening in this system, as presented in different places in the text: outward pressure from the expanding oocyte, adhesion between follicle cells and the lengthening oocyte, and potential forces from the actomyosin networks themselves. It's sometimes unclear how separable these sources are, which makes it difficult to evaluate the claim that actomyosin networks drive cell expansion in the system. This is something that I think is worth addressing in the Discussion.

Answer: We thank the reviewer for pointing out this unclear point in the paper. As suggested by the reviewer, we now discuss in the text how different sources are involved in follicle cell flattening, how they coordinate with each other, and how actomyosin potential forces drive cell expansion with the participation of other potential factors. This discussion in the text (page 18-19, yellow label) is as follows:

“Regarding follicle cell flattening during *Drosophila* late oogenesis, our study reveals multiple potential causes ranging from internal to external regions of the tissue, including: outward pressure from the expanding oocyte, adhesions between follicle cells and the elongating oocyte, and potential forces from the actomyosin networks themselves. Based on our results, we propose some unique contributions of these 3 potential causes. 1) Outward pressure from the expanding oocyte is the major driver of follicle cell flattening process, which could be indirectly reflected by the loss of follicle cell spreading behaviour if dumping-induced oocyte growth is inhibited. 2) In response to oocyte outward pressure, adhesions between follicle cells and the oocyte acts as a bridge between the increasing apical surface of follicle cells and the surface of the growing oocyte surface; correspondingly, follicle cells gradually flatten to accommodate the increasing oocyte surface. 3) During this flattening process, follicle cells gradually enlarge not only their apical surface but also their basal surface to maintain the normal epithelial cell shape; but unlike cell-cell adhesions at cell apical surface, follicle cells interact with the ECM at their basal surface²⁴, so the expanding properties of actomyosin networks proposed in this study could stabilize or reinforce cell basal expansion when the cell basal surface increases during cell spreading on the ECM. Here, we regard these unique contributions of different potential causes as a series of mechanical changes from internal to external tissue regions (e.g. internal pressure transfer from the oocyte to follicle cells might lead to Rac1 upregulation and Rho1 downregulation in follicle cells for their ruffling behaviour); but at the same time, external mechanical responses could in turn modulate internal tissue changes⁴⁴, such as oocyte growth and elongation, indicating that E-cadherin adhesions between follicle cells and the oocyte act as a bridge mediating some feedback mechanisms to maintain structural integrity. Therefore, even with unique contributions, these potential causes are not completely separable but are highly integrated to direct follicle cell flattening and cell/oocyte changes. This interdependence has been reflected in the fact that if oocyte growth or actomyosin network expansion property is modulated, some corresponding changes occur in follicle cells or the oocyte. Thus, our observed coordination between follicle cells and the oocyte might be similar to the situation encountered during late elongation in *C. elegans*, which depends on a mechanical crosstalk between muscles and epidermis⁴⁵. Future studies are needed to clearly understand different unique contributions as well as their interdependence for follicle cell flattening process, as well as the mechanistic feedback

mechanisms of follicle cells and the oocyte to achieve their tight coordination.”

2. The simulation results in which Rho1 inhibition only inhibits P-A migration when the oocyte directional growth reduction is included are interesting but raise questions regarding the role of Rho1 in cell spreading. Do the authors have an argument for why Rho1 reduction alone is not sufficient to reproduce the experimental results? Does Rho1 inhibition in the follicle cells somehow inhibit nurse cell dumping, which then inhibits follicle cell expansion?

Answer: Indeed, oocyte growth and the spreading of follicle cell monolayer are physically closely linked. Follicle cell spreading requires sufficient free area (through oocyte growth) as a prerequisite. Alternatingly increasing free area and follicle cell spreading area are necessary to coordinate normal A-P extension of the egg chamber. Thus, it is easy to understand that Rho1 inhibition represses P-to-A migration only when oocyte growth is reduced and free area is limited.

As for why reducing Rho1 alone is not enough to reproduce the experimental results? We propose that the driving force and prerequisite for follicle cell active migration is Rac1 which induces the protrusions of the leading edge; and a certain amount of active Rho1 can effectively promote the formation of stress fibers and mature focal adhesions as a complementary way to support follicle cell active migration induced mainly by Rac1. Thus, if the passive migration mode (oocyte growth induced by dumping) is not taken into account, Rho1 inhibition has a limited effect on follicle cell expansion, which is reflected by our single-factor simulation.

As for whether inhibiting Rho1 in follicle cells can inhibit nurse cell dumping and follicle cell expansion? From our experimental results, we observed that the inhibition of Rho1 led to abnormal (strong slow-down) oocyte elongation and follicle cell spreading. We propose that, on one hand, Rho1 inhibition can decrease active migration of follicle cells by the regulation of actomyosin and other related signaling. On the other hand, some mechanical feedback mechanisms might exist to modulate nurse cell dumping and directed oocyte growth by Rho1: e.g. Rho1 inhibition can allow the egg chamber to grow not only along the A-P axis but also along the D-V axis; consequently, oocyte A-P extension is suppressed while D-V extension is increased. This modulation of directed oocyte growth might in turn indirectly suppress follicle cell spreading. Thus, while there is no conclusive evidence as to whether Rac1DN or Rho1DN (in follicle cells) have a direct effect on dumping at this time, our simulations suggest that the disturbance of dumping can greatly affect the A-P elongation of the egg chamber.

3. In general, the text did not guide the reader through each figure as well as it could have. For Figure 1, the figure callouts to 6-9 figure panels at a time were confusing because the authors didn't explain what each measurement was and why it was important. For Figure 5, there were similar massive callouts without explanation Lines 252-255. I think the paper would be more easily consumable if the authors provided more guidance for their analysis.

Answer: We thank the reviewer for pointing out this issue. Now we include the detailed information of Figure 1 and Figure 5 in the text to explain what each measurement was and why it was important. The detailed information of Figure 1 in the text (page 5, yellow label) is as follows:

"Here, we used dynamic myosin intensity to quantify basal myosin pulsatile behaviour (Fig. 1c), relative myosin and F-actin intensity to quantify the strength of basal actomyosin networks (Fig. 1d, e), and the segmentations and distribution angles to quantify stress fiber polarity (Supplementary Fig. 1e, f), in order to show the gradual modulation of basal actomyosin pulsations during different

stages. Meanwhile, we used circularity index to quantify cell ruffling behaviour (Fig. 1f), ruffle direction to quantify where ruffles occur within the cell (Fig. 1g), the area and perimeter fold change to quantify the increase in basal cell area and perimeter (Fig. 1h), and basal area and velocity to quantify cell basal expansion and migration processes (Fig. 1j, j); and these different factors demonstrate various cell behaviours including ruffling, increasing and spreading processes.”

The detailed information of Figure 5 in the text (page 12, yellow label) is as follows:

“Here, we monitored cell ruffles in the anterior, middle and posterior regions of the follicular epithelium during the pre-wave vs. ruffling phases (Fig. 5d); we then quantified the circularity index dynamics of cells in different tissue regions (Fig. 5e, g, i); finally, we acquired the temporal correlation of ruffling occurrence in different tissue regions (Fig. 5f, h, j). From this temporal correlation analysis, we confirmed that ruffling process in WT tissues began in tissue posterior region and spread to more anterior region (Fig. 5d-f). Differently, dumping-deficient tissues sustained follicle cell ruffling process (Fig. 5d), while their ruffling behaviour started simultaneously throughout the tissue, losing P-to-A waves (Fig. 5d, g-j). Then, we characterized the spreading behaviour of follicle cells by quantifying the A-P migration distance and velocity of follicle cells. Importantly, compared with WT tissues, dumping-deficient tissues strongly lost follicle cell spreading process (Fig. 5k-m), consistent with their defective oocyte growth.”

Minor comments:

1. Fig. 1c: colors for Increasing and Ruffling are reversed relative to the rest of the figure panels.

Answer: We thank the reviewer for pointing out this mistake. In the updated Fig. 1c, we correct this mistake, as shown below.

2. “our results support Rac1 and Rho1 signalling as key factors controlling the ruffling and spreading process, respectively, to enlarge cell basal surface area”: it seems Rac1 is also important for spreading (Fig. 3i), It seems that the authors should not conclude so strongly that the two GTPases have such cleanly separate activities here.

Answer: We thank the reviewer for pointing out this overstatement part. Thus, in the updated text (page 9, yellow label), we change the previous sentence into: “our results support that Rac1 signaling is a key factor in controlling the ruffling and spreading processes and Rho1 signaling is a key factor in controlling the spreading phase in order to expand the cell basal surface area”.

3. Lines176-178, The defect in basal area could also reflect a defect in migration.

Answer: We thank the reviewer for pointing out this inaccurate sentence. Now we change this part into:

“a severe defect in basal area increase during the increasing phase (Fig. 3h, i), supporting ruffles as a membrane reservoir to accommodate subsequent drastic increase in basal area (Fig. 1h); 3) blockade of cell spreading process (Fig. 3j, k), implicating the importance of ruffles for collective

movement over ECM as well as its further role in increasing basal area during the spreading phase (Fig. 3i).”

This is updated in the text (page 8-9, yellow label).

4. Lines 183-186: Need the word “respectively” added. E.g. “ promoted or inhibited collective cell spreading process, respectively, . . .”

Answer: We now add the word “respectively” to this sentence, as the reviewer suggested (page 9, yellow label).

5. Line 207-8: This sentence was confusing, particularly the term ‘focal adhesion continuum’

Answer: we thank the reviewer for pointing out this confusing sentence. Now we change ‘focal adhesion continuum’ into “focal adhesions and their associated basal stress fibers” in our text (page 8, yellow label; page 10, yellow label).

6. Lines 199-200: How does ‘enhanced focal adhesions’ translate into the stability of focal adhesions? The authors show that there can be two different focal adhesion states – one static and one more dynamic.

Answer: Our proposed static/dynamic focal adhesion states mean that neighboring focal adhesions do not have or have a chronological order of the turnover. Here, we have no conclusive evidence that the static/dynamic state is due to the stability of focal adhesions. Thus, we cannot conclude that enhanced focal adhesions can translate into either more or less stability of focal adhesions. Differently, we feel that focal adhesion link with contractile or expanding stress fibers should be the major factor controlling these two different states, while enhanced or reduced focal adhesion levels just modulate the adhesion interactions between cells and the matrix (such as more or less focal adhesions locally in contact with the matrix). Thus, enhanced focal adhesions associated with different stress fibers could produce two distinct phenotypes: either more reducing effects on basal area of follicle cells, or faster spreading effects of follicle cells on the matrix.

7. Fig. 4b: are Increasing and Spreading data reversed in the plot or in the legend?

Answer: We thank the reviewer for pointing out this mistake. In the updated Fig. 4b, we correct this mistake, as shown below.

8. Lines 230-233: Can the authors explain why Rac1 inhibition causes significantly delayed cell flattening while only moderately repressing oocyte A-P directed growth?

Answer: We propose that the rapid flattening process (which occurs during the increasing phase) requires membrane pre-accumulation during the ruffling phase. Rac1 inhibition results in reduced ruffle duration or/and dynamics, leading to insufficient membrane pre-accumulation. Delayed cell flattening caused by Rac1 inhibition demonstrates the necessity of Rac1 in ruffling.

However, due to the important role of Rac1 in the rapid flattening process, many tissues with strong Rac1 inhibition were unable to grow to the spreading stage, so we cannot use these tissues to study the effects of Rac1 inhibition on the oocyte A-P directed growth. Here, tissues with weak Rac1 inhibition, allowing follicle cells to enter the spreading process and allowing oocytes to grow, have been selected for our analysis, thereby leading to moderate repression of oocyte A-P directed growth.

9. Fig. 3c caption: would “Paxillin-overexpressing” instead of “paxillin-expressing” be more accurate?

Answer: We correct this word in the text (page 27, yellow label), as suggested by the reviewer.

10. Fig. 4j: presenting the outlines with different maximum elapsed time is slightly confusing. Could the authors either extend the times used for Pax OE and Talin RNAi to 100 min or explain why those times cut off at 60 min?

Answer: We set the elapsed time to 60 minutes for Pax-OE because its oocytes grow faster, completing within 60 minutes. Correspondingly, we also set the elapsed time of Talin RNAi-expressing egg chamber to 60 minutes. We apologized for the confusion.

Now in the updated Fig. 4j, we extend the elapsed time of Talin RNAi to 100 minutes. In order to better demonstrate the accelerated effect of Pax-OE on oocyte growth, we extend the elapsed time of Pax-OE to 70 minutes, which clearly indicates that oocyte growth is completed earlier in the Pax-OE egg chamber.

11. Fig. 6a, b: Is myosin intensity equivalent or different between the phases?

Answer: In Fig. 6a-b, the myosin-2 intensity is equivalent between the pre-wave and spreading phases, as demonstrated by the quantification shown in Fig. 1d.

12. Line 291: Sentence requires “respectively” E.g. “pre-wave or spreading phase, respectively, consistent with . . .”

Answer: We now add the word “respectively” to this sentence (page 14, yellow label), as the reviewer suggested.

13. Line 295-296: Just 1 sentence should not be a paragraph.

Answer: We correct this word as suggested by the reviewer.

14. Acknowledgements: “Jennifer Zallet” should be “Jennifer Zallen”

Answer: We thank the reviewer for pointing out this mistake. Actually, in the acknowledgements, we misspelled the name of Jennifer Zanet, who previously worked on Fascin (Zanet J et al., J Cell Biol [2012] 197(4):477-86).

To make it easier for reviewers to track our changes, we mark new insertions in yellow and deletions as strikethroughs.

Reviewer #3 (Remarks to the Author):

Major comments:

1. The temporal model of Fig. 2p, along with the text, suggests that Rac signaling is the first key event. However, unless mistaken, Rho signaling is strong until stage 10 to stimulate the basal Myo2 pulses. So, I would have imagined that the first key event once dumping starts should be Rho down-regulation, which the authors did not consider. If however it is Rac1 that induces RhoA downregulation, how does this occur? It would be useful to cross-examine RhoGAPs and RacGEFs and their activity at the stage 10 transition. The Bellaïche lab very recently published a systematic study of all Rho/Rac GAPs and GEFs present in follicle cells which should be useful.

Answer: We thank the reviewer for pointing out this interesting question. We first discussed with Yohanns Bellaïche: considering the recent high volume of fly request, his laboratory can only support some of these GEF/GAPs in a short period of time. Then, based on our observed phenomenological features and the published subcellular patterns (in Bellaïche's paper), we chose some potentially important RacGEFs/RhoGAPs for their cross-examination. Our conclusions and Rebuttal Fig. 4 are as follow:

About the potential RacGEF, we identified three major categories:

1) Cdep-GFP showed similar dynamic expression and basal distribution to the Rac1 activity reporter (PAK3-RBDGFP) during follicle cell wave behaviours (Rebuttal Fig. 3a, left panels), thereby implicating that Cdep is mostly likely the key RacGEF in follicle cells. The strong increase in Cdep-GFP at follicle cell basal domain seems to be consistent with the strong increase in PAK3-RBDGFP during ruffling (Rebuttal Fig. 3a, left panels). Although this result might suggest that Rac1 strong activation could induce the Rho1 downregulation, our genetic inhibition of Rac1 or downstream effectors (Wave/Scar complex) didn't show strong restoration of basal stress fibers (see Supplementary Movie 4). Thus, we speculate that there may be a more complex signaling crosstalk between Rac1 and Rho1 signaling rather than direct mutual antagonism, meaning that that Rac1 activation might not be required to down-regulate Rho1 signaling, or vice versa. Indeed, we found that Rac1 and Rho1 activities synergize to positively regulate actomyosin networks at supracellular cables of migrating border cells (Zhou et al., Nat Commun 2022, 13(1):6014) as well as pulsatile basal actomyosin networks in follicle cells during stages 9-10 (our another prepared manuscript).

2) Three potential RacGEFs (Vav, Mbc and Sif) didn't show prominent signals at follicle cell basal domains before and during follicle cell wave behaviours (Rebuttal Fig. 3c, left panels), thereby excluding them as key RacGEFs controlling follicle cell expansion waves.

3) Unexpectedly, RtGEF (a potential Rac1/Cdc42 GEF) showed strong distribution patterns at follicle cell basal domains, especially at focal adhesion regions (two sides of D-V distributed stress fibers), during the pre-wave and after-wave phases; but these RtGEF basal signals completely disappeared in wave-occurring follicle cells while being maintained in more anterior follicle cells before the wave arrival (Rebuttal Fig. 3b). Regarding its distribution pattern as well as its negative correlation to wave occurrence, we speculate that RtGEF might be a key RhoGEF controlling contractile stress fibers, which needs to be reduced to allow the formation of expanding stress fibers, while implicating the presence of another RhoGEF controlling expanding stress fibers. However, since we didn't request any other RhoGEF, we are currently unable to identify another RhoGEF as critical for controlling the formation and maintenance of expanding stress fibers during epithelial

cell expansion waves.

About the potential RhoGEF, we identified two major categories:

1) RhoGap19D showed prominent dynamic expression and distribution in follicle cell basal domain during follicle cell wave behaviours (Rebuttal Fig. 3a, right panels). Although we detected a strong increase of RhoGAP19D during ruffling, this RhoGAP19D showed even stronger basal signals during spreading phase (Rebuttal Fig. 3a, right panels). This gradual increase in basal RhoGAP19D signals implicates that it is a key RhoGAP for wave initiation and progression. But its pattern was not consistent with Rho1 activity reporter (Ani-RBDGFP), meaning: either another unidentified RhoGAP might be reduced during spreading phase, or expanding stress fibers might cooperate with above-mentioned unidentified RhoGEF to control pulsatile Rho1 activity and expanding stress fibers. Here, we also speculate that there may also be more complex signaling crosstalk between RhoGEFs (potentially two functionally opposite RhoGEFs) and RhoGAP to control the reduction of contractile stress fiber pulses and the formation/maintenance of expanding stress fiber pulses.

2) Three potential RacGEFs (RhoGAP1A, RhoGAP5A and RhoGAP15B) didn't show prominent signals at follicle cell basal domains before and during follicle cell wave behaviours (Rebuttal Fig. 3c, right panels), thereby excluding them as key RhoGAPs controlling follicle cell expansion waves. Our hypothesized complex signaling crosstalk seems to be also supported by unpublished work from Eurico Morais-de-Sá (another corresponding author in the Bellaiche's paper). They informed us of the presence and roles of RhoGAP1A, RhoGAP19D and RhoGAP15B in controlling myosin activity in border cells, as well as the complexity of all GEFs/GAPs (not direct mutual antagonism between RacGEF and RhoGAP) in both migrating border cells as well as follicle cells before stage 9. Thus, it is likely that this complexity is also present in follicle cells during late oogenesis, such as epithelial cell expansion waves (including the stage 10 transition). A full understanding of RacGEF/RhoGAPs (and possibly the two functionally opposite RhoGEFs) will be a huge and time-consuming research effort, which would very likely result in another complete publication story. Thus, considering the pre-mature RacGEFs/GAPs results shown here, we decide not to present these interesting results in our current revision.

Rebuttal Figure 3. Potential RacGEFs and RhoGAPs during different phases of follicle cell expansion waves.

a) Representative images and summary cartoons of Cdep-GFP (RacGEF, in left panels) and

RhoGAP19D-GFP (RhoGAP, in right panels), as well as LifeAct-RFP (F-actin reporter), in follicle cell basal domains during 3 indicated phases. Heatmap is used to better view relative activity of Cdep-GFP and RhoGAP19D-GFP in follicle cells. **b)** Left: representative cartoon summarizing follicle cells with the wave occurrence or before the wave arrival; Right: representative images of RtGEF (potential RhoGEF) and LifeAct-RFP in follicle cell basal domains during 4 indicated phases. White dotted rectangles mark enlarge images showing that RtGEF-GFP signals are specifically distributed at both ends of basal stress fibers, supporting that RtGEF might be a key RhoGEF controlling contractile stress fibers in follicle cells. **c)** Representative images and summary cartoons of 3 other RacGEFs (Mbc, Sif and Vav, in left panels) and 3 other potential RhoGAPs (RhoGAP1A, RhoGAP5A and RhoGAP15B, in right panels), as well as LifeAct-RFP (F-actin reporter), in follicle cell basal domains during 3 indicated phases. Scale bars are 10 microns in all figures.

2. Related to the issue of the respective and mutual mechanical interactions between the oocyte and follicle cells, the authors should briefly speculate on the signaling process leading to Rac1 and RhoA activity up- or down-regulation, as well as to the signaling preventing oocyte expansion when follicle ruffling or spreading is hampered. At the mesoscopic level, but not at the signaling and cellular levels, it is reminiscent of the situation encountered during *C. elegans* late elongation which depends on a mechanical cross-talk between muscles and the epidermis.

Answer: We thank the reviewer for pointing out this interesting question. But due to the limited understanding of the upstream GEFs/GAPs controlling the Rac1 upregulation and the Rho1 downregulation, it is difficult for us to speculate on the signaling process for respective changes in Rac1 and Rho1. This is the same case about the signaling/signals preventing oocyte expansion when follicle ruffling or spreading is hampered. Considering that reviewer 2 asked a similar question, we thus include very brief speculations for the Rac1/Rho1 changes in follicle cells as well as how ruffling/spreading hamper prevents the oocyte expansion, in the text (page 18-19, yellow label), and also as follows:

“...Here, we regard these unique contributions of different potential causes as a series of mechanical changes from internal to external tissue regions (e.g. internal pressure transfer from the oocyte to follicle cells might lead to Rac1 upregulation and Rho1 downregulation in follicle cells for their ruffling behaviour); but at the same time, external mechanical responses could in turn modulate internal tissue changes ⁴⁴, such as oocyte growth and elongation, indicating that E-cadherin adhesions between follicle cells and the oocyte act as a bridge mediating some feedback mechanisms to maintain structural integrity...”

We also enjoyed the reviewer’s comments about the comparison between *C. elegans* elongation and our system. Our comparison part is as follows:

In the *C. elegans* elongation, after undergoing a first elongation driven by the contraction of epidermal cells, during the second elongation stage, actomyosin-driving forces of seam cells are no longer sufficient to propel further squeezing, and the underlying musculature becomes the main force generator (ref: Carvalho, C.A. & Broday, L. Game of Tissues: How the Epidermis Thrones *C. elegans* Shape. *J Dev Biol* 8 (2020)). *C. elegans* elongation and follicle cell expansion present some major differences, including multilayer structure and interaction, force generators and actomyosin types. But both systems support that the cell-cell coordination associated with mechanical feedback

is necessary for the normalization of tissue elongation. However, due to the limited information about the mechanical feedback between follicle cells and the oocyte, we only briefly discuss this point in our updated discussion (page 19, yellow label), and also as follows:

“...Therefore, even with unique contributions, these potential causes are not completely separable but are highly integrated to direct follicle cell flattening and cell/oocyte changes. This interdependence has been reflected in the fact that if oocyte growth or actomyosin network expansion property is modulated, some corresponding changes occur in follicle cells or the oocyte. Thus, our observed coordination between follicle cells and the oocyte might be similar to the situation encountered during late elongation in *C. elegans*, which depends on a mechanical crosstalk between muscles and epidermis⁴⁵. Future studies are needed to clearly understand different unique contributions as well as their interdependence for follicle cell flattening process, as well as the mechanistic feedback mechanisms of follicle cells and the oocyte to achieve their tight coordination.”

3. The presentation of the final model should be improved. For readers who would not want to go through the Methods section, it is not easy to understand what was done just reading the main text and Fig. 7 legend. Please indicate

Answer: As suggested by the reviewer, we now inform all following points in the text (page 32, yellow label).

- what are the red lines in 7c & their respective intensity in 7d (how do they relate to stress fibers);

Answer: The red lines in figure 7c and 7d represent stress fibers along the DV direction in increasing and spreading phases. The thickness of stress fibers represents the relative intensity of stress fibers: the stress fibers of Rho1 DN and Talin RNAi are reduced, while the stress fibers of Pax OE are increased. Now we update the detailed information in the Figure and Movie legends.

- what are the graded grey rectangles in 7e, which are unchanged in all conditions except in dumping

Answer: The graded grey rectangles represent the propagation of expansion waves from posterior to anterior region in different genetic backgrounds. In most genetic backgrounds, we can detect a gradient of wave occurrence, whereas in dumping defective background, this gradient is disturbed thereby implying the loss of expansion wave. Now we update the detailed information in the Figure legend.

- describe also briefly the speedometer

Answer: Speedometers are used to demonstrate the cell migration speed in different genetic backgrounds. Now we update the information in the Figure legend.

- it is not clear what has been combined in Fig. S11

Answer: Now we include the following information in the Supplementary information (page 15, yellow label):

“To further explore the passive drive of follicle cell expansion, the effect of oocyte A-P extension was considered in the combined-factor simulation.”

- the comparison between the single-factor simulations and the actual experimental data do not entirely match for most parameters either in absolute terms (as they point out for Rho1DN) or in relative terms; this should be better commented. It might for instance indicate that Rac1 influences RhoA and vice-versa.

Answer: Deviations between single-factor simulation and experimental data can be due to a number of factors: e.g. 1. Optimized vertex models are still not enough to perfectly replicate follicle cell ruffling. 2. We do not include multifactorial interactions. In single-factor simulations, we didn't

capture the strong inhibition of cell migration in Rho1 DN condition as in experiments.

To explain why Rho1DN in single-factor simulation cannot perfectly produce strong inhibition effect, we include more detailed information in the text, as follows:

“To note, in single-factor simulations we pre-set identical oocyte A-P extension (also used in other genetic backgrounds), whereas in experiments Rho1DN has strongly inhibited oocyte extension (also observed in other genetic backgrounds). Thus, we propose that the difference is due to our neglect of the effect of oocyte A-P extension on cell migration, suggesting the existence of a passive migration due to oocyte A-P extension besides active migration induced by cell protrusions controlled by Rac1, Rho1 and focal adhesions. To confirm our hypothesis, we introduced different oocyte A-P extension in different genetic backgrounds as observed in experiments, which we called the combined-factor simulation (Supplementary Fig. 11 and Supplementary Movie 14; see Methods). The combined-factor simulations showed that the inhibition of Rho1 and the simultaneous defect in oocyte A-P extension severely hindered collective cell migration (Fig. 7d and Supplementary. 11f, g), thus emphasizing that oocyte directional growth by nurse cell dumping is critical for driving passive migration mode.” (in our text page 15-16, yellow label)

Minor:

1. Author contributions: the contributions of Jiaying Liu (JL), Kaifu Yang (KY), Vanessa Dougados (VD) and Thomas Mangeat (TM) have not been listed in the relevant section.

Answer: We input the missing information in the text (page 52, yellow label), as pointed out by the reviewer.

2. Line 348-354: the words adaption and adaptation were used indifferently; it would be better to stick to a single word.

Answer: We only used “adaptation” in Line 348-354, as the reviewer suggested. It is updated in the text (several word replacements in page 20-21, yellow label).

3. Fig. 1 legend: the duration of each phase as indicated in the main text (lines 94-100) or in panel bright are not the same; please clarify.

Answer: We thank the reviewer for pinpointing this confusing part in our text. The duration of each phase indicated in the text is the total period of each phase. To better display the morphological dynamics of follicle cell basal domains during these 4 phases, a shorter window of time was selected, especially during spreading phase. Thus, we now include a description of the elapsed time in Fig. 1b legend (page 23, yellow label) to minimize this confusing point: “in selected periods of each phase”.

4. Fig. 2bc legend: are fold changes measured over the entire area of the cell or over a small subsection? Please clarify here and in the methods section.

Answer: Thank the reviewer for pinpointing this confusion point in Fig. 2b-c legend. Here, area changes refer to the subcellular area associated with membrane ruffling. As the reviewer suggested, we clarify the detailed information in Fig. 2b-c legend (page 25, yellow label) and methods section (Page 42, yellow label).

5. Fig. 2pq legend: what is the time unit in those panels?

Answer: The time intervals depicted in the cartoon (Fig. 2p-q) are only relative and intended to clarify the chronology of the overall events.

6. Fig. 2e,h: the respective peaks of actin versus Rac1 or Rho1 differ by about 1 min, which is much higher than those reported in the germband or in early *C. elegans* embryos (in the range of a few seconds). Could you comment.

Answer: The pulsatile system (Rac1, Rho1 and stress fiber F-actin) during epithelial cell expansion waves is similar to basal myosin oscillation during stage 9 and 10B reported in our previous study (Qin X et al., Nat Commun [2018] 9(1):1210). The cycle period is around 6-8 minutes and the delay between Rock and myosin pulses is close to 1 minute (Qin X et al., Nat Commun [2018] 9(1):1210). Both the cycle period and the signal delay period are much longer than other pulsatile systems reported in the germband or in early *C. elegans* embryos.

7. Fig. 1c versus Fig. 2b, e, h, k, p: it is not easy to juxtapose the myosin-2 fluctuations from Fig. 1c with those of actin in Fig. 2, despite the fact that Fig. S1 shows that actin and myosin-2 are fully cross-correlated.

Answer: Myosin-2 pulses (in Fig. 1c) and F-actin/myosin-2 pulses (Fig. S1) are total intensities from whole cell basal domain, whereas F-actin pulses in Fig. 2b, e, k are the respective intensities from subcellular basal regions (associated with either random ruffles during ruffling phase or leading ruffles during spreading phase). This signal difference between total and sub region of basal domain might result in the inconsistency between Fig. 1c and Fig. 2b, e, k.

Differently, F-actin pulses in Fig. 2h, n are total intensity from whole cell basal domain, the same as in Fig. 1c and Fig. S1. This inconsistency is because actomyosin pulses in different cells and time periods vary greatly in periodicity, as demonstrated below in Rebuttal Fig. 4:

Rebuttal Figure 4. Irregularity of basal actomyosin pulses in follicle cells during different stages.

a) Comparison of the oscillation patterns used in Fig. 1c and Fig. 2h/n. **b)** Quantification of myosin/F-actin cycle periods in follicle cells during 4 indicated phases (showing a big variation of actomyosin cycle periods during the spreading phase).

8. Fig. 4b: define the “flatten ratio” (line 470)

Answer: As the reviewer requested, we define the “flatten ratio” in Fig. 4b legend (page 28, yellow label): “Here, flatten ratio is used to define the status of follicle cell flattening process, and its quantification is shown in method section”. Indeed, this calculation of flatten ratio has previously been listed in the Method section (page 42).

REVIEWER COMMENTS

Reviewer #1 (Remarks to the Author):

The authors have carefully addressed my previous questions. The inclusion of the new data and discussion are very helpful. I now fully support the publication of this elegant piece of work in Nature Communications.

Below are some minor points regarding the revision:

1. The new data presented in the point-by-point response letter showing the effects of nurse cell dumping on Rac1 and Rho1 activity is very interesting. I agree with the authors that the mechanism of the observed effect merits further investigation in follow-up studies and therefore it is not necessary to include the data in the current paper.
2. The RGB color mark in Supplementary Figure 2j is a little confusing. The drawing seems to indicate that cells in region A, B and C initiate the ruffling phase sequentially, but enter the increasing phase and spreading phase at the same time. However, in the text, the authors state that "Tissue-scale analysis revealed that follicle cell ruffling and spreading processes initiated posteriorly and eventually propagated towards the anterior".
3. Line 464: "For example, experiencing ruffling makes follicle cells more susceptible to metamorphosis". It is a little unclear why metamorphosis is mentioned in this context. The authors have carefully addressed my previous questions. The inclusion of the new data and discussion are very helpful. I now fully support the publication of this elegant piece of work in Nature Communications.

Below are some minor points regarding the revision:

1. The new data presented in the point-by-point response letter showing the effects of nurse cell dumping on Rac1 and Rho1 activity is very interesting. I agree with the authors that the mechanism of the observed effect merits further investigation in follow-up studies and therefore it is not necessary to include the data in the current paper.
2. The RGB color mark in Supplementary Figure 2j is a little confusing. The drawing seems to indicate that cells in region A, B and C initiate the ruffling phase sequentially, but enter the increasing phase and spreading phase at the same time. However, in the text, the authors state that "Tissue-scale analysis revealed that follicle cell ruffling and spreading processes initiated posteriorly and eventually propagated towards the anterior".
3. Line 464: "For example, experiencing ruffling makes follicle cells more susceptible to metamorphosis". It is a little unclear why metamorphosis is mentioned in this context.

Reviewer #2 (Remarks to the Author):

The authors have addressed most of my specific comments, with the exception of improving the clarity and readability of the manuscript. I am an expert in the cytoskeleton and morphogenesis field and it took me hours trying to evaluate the authors plots and whether they properly support the conclusions and sometimes I was not seeing at all what they were

talking about at all. The manuscript in its current state will not be broadly accessible to the readers of Nature Communications, which would be unfortunate because I think it is a nice study that shows a new mechanism for cell flattening. I will give some specific examples, but these examples reflect a broader problem throughout the entire manuscript. It requires careful editing and a logical and consistent ordering of explanations of figure panels for readers to not feel lost.

Main comments:

Lines 97 – 127: I appreciate that the authors have mentioned what each of the plots are now, but what readers need is for the authors to explain what specific measurements are enabling the authors to distinguish between the different phases. Making a statement and then calling out no less than 6 figure panels is not helpful. The authors need to direct the reader to the most important plot.

Lines 149 – 162: Author are going back and forth between different phases. And all of this is in a behemoth of a Supplemental figure where some panels have some phases quantified and not others. The whole section on Rac1 and Rho1 activity was incredibly confusing and was very disjointed with ideas coming from all over the place. The message should be consistent.

Line 253: Reference to Fig. 4d. I can't see how 4d corresponds to the statement in the sentence.

Lines 284 – 286: Reference to 5d, g-j. I don't know what I am looking at. The authors need to explain how these plots are leading them to make the stated conclusion.

Lines 320-322: Reference to Fig. 6f, g. Again, the authors need to point me towards what I am looking for in these figures.

Line 334, 384, 389. Reference to expanding structures and stress fibers. There is no evidence that a myosin-containing structure is expanding. What is more likely happening, is that myosin fluidizes the cortical network, leading to turnover that allows 'cell expansion' through stress dissipation. I would point to other studies that have shown the role of myosin in actin turnover and the role of actin turnover in stress dissipation (PMID: 30538127).

Line 363-366: Reference to Supp Fig 11. Just say no to calling out a figure with 9 panels. Explain how what is in the figure supports your statement.

Minor comments:

Lines 25, 50: 'junction' should be 'junctions'

Line 105: 1j, j should be 1j

Line 194: 'blockade of cell spreading process' should be 'blockade of the cell spreading process'

Reviewer #3 (Remarks to the Author):

The authors have done a commanding job at addressing reviewers' comments. Their revised discussion is overall clearer and more balanced. Regarding their detailed responses, I agree that identifying the RhoGAP acting to dampen RhoA early ones beyond the scope of this manuscript if there are multiple GAPs and GEFs involved, but it is really good that they initiated that search. This said, if the comment on the fact that there might be a need to dampen RhoA activity to initiate the process, they could mention the issue in their discussion towards lines 475. I would also recommend that they include the 1st "Rebuttal Figure 1" as a supplementary figure in the final manuscript, and potentially the 4th Rebuttal Figure. I now fully support publication in Nature Communications.

To make it easier for reviewers to track our changes, we mark new insertions in yellow and deletions as strikethroughs.

Reviewer #1 (Remarks to the Author):

Answer: We thank the reviewer for improving our manuscript.

1. The new data presented in the point-by-point response letter showing the effects of nurse cell dumping on Rac1 and Rho1 activity is very interesting. I agree with the authors that the mechanism of the observed effect merits further investigation in follow-up studies and therefore it is not necessary to include the data in the current paper.

Answer: We agree with the reviewer and do not include the data in the current paper.

2. The RGB color mark in Supplementary Figure 2j is a little confusing. The drawing seems to indicate that cells in region A, B and C initiate the ruffling phase sequentially, but enter the increasing phase and spreading phase at the same time. However, in the text, the authors state that “Tissue-scale analysis revealed that follicle cell ruffling and spreading processes initiated posteriorly and eventually propagated towards the anterior”.

Answer: We thank the reviewer for pointing out this confusion sentence. Here, we consider ruffling and spreading processes as an integrated "wave" behavior: temporally, cells in region A, B and C initiate the ruffling phase sequentially, but enter the increasing and spreading phases at the same time; spatially, ruffling occurrence gradually expands from posterior to anterior, as well as spreading process expanding from posterior to anterior, thereby showing the global wave behavior from posterior to anterior.

To clarify this confusing sentence, we revise our original sentence into:

“Tissue-scale analysis revealed that follicle cell expansion behavior initiated posteriorly and eventually propagated towards the anterior (Fig. 1k, l and Supplementary Fig. 2h-j): temporally, posterior follicle cells enter the ruffling phase sequentially, but simultaneously enter the increasing and spreading phases; spatially, both ruffling and spreading processes progressively expand from posterior to anterior (Supplementary Fig. 2j).” This is updated in (page 6, yellow label).

3. Line 464: “For example, experiencing ruffling makes follicle cells more susceptible to metamorphosis”. It is a little unclear why metamorphosis is mentioned in this context.

Answer: We thank the reviewer for pointing out this confusion word. Now, we update “metamorphosis” into “plastic (more prone to morphological changes)” to make our meaning clear. This change is updated in (page 21, yellow label).

To make it easier for reviewers to track our changes, we mark new insertions in yellow and deletions as strikethroughs.

Reviewer #2 (Remarks to the Author):

Answer: We thank the reviewer for improving our manuscript.

1. Lines 97 – 127: I appreciate that the authors have mentioned what each of the plots are now, but what readers need is for the authors to explain what specific measurements are enabling the authors to distinguish between the different phases. Making a statement and then calling out no less than 6 figure panels is not helpful. The authors need to direct the reader to the most important plot.

Answer: We thank the reviewer for pointing out this unclear section. Now we include more information and the corresponding figures to direct the reader for different phases and cellular behaviors. These changes are updated in (Page 5, yellow labels).

2. Lines 149 – 162: Author are going back and forth between different phases. And all of this is in a behemoth of a Supplemental figure where some panels have some phases quantified and not others. The whole section on Rac1 and Rho1 activity was incredibly confusing and was very disjointed with ideas coming from all over the place. The message should be consistent.

Answer: As the reviewer requested, we now revise this whole section based on each phase to compare the activity (and distribution) of Rac1, Rho1 and focal adhesions. We also include the quantification of Rho1 activity during increasing phase (current Supplementary Fig. 4f), so that all Rac1, Rho1 and focal adhesions have been compared for all 4 phases. To make the figure organization clearer, we move the original Supple Fig. 4e-f to current Supple Fig. 4o-q, and all these changes are updated in our current text (results, figure legends and supplementary note 1). The major section of these changes is updated in (Pages 7-8, yellow labels).

3. Line 253: Reference to Fig. 4d. I can't see how 4d corresponds to the statement in the sentence.

Answer: We thank the reviewer for pointing out this unclear section. Since follicle cell boundary outlines from 0 to 100 minutes blocked us to precisely view cell shape during the spreading process, boundary outlines at 0 and 100 minutes have been shown separately to highlight follicle cell elongation status, as shown below:

Now we update both figure (in Fig. 4d) and this information (in Fig. 4d legend; Page 29, yellow labels).

4. Lines 284 – 286: Reference to 5d, g-j. I don't know what I am looking at. The authors need to explain how these plots are leading them to make the stated conclusion.

Answer: We thank the reviewer for pointing out this unclear section. Now we revise this section and include the information to explain how we can conclude from these figures (Page 13, yellow labels).

5. Lines 320-322: Reference to Fig. 6f, g. Again, the authors need to point me towards what I am looking for in these figures.

Answer: We thank the reviewer for pointing out this unclear section. Now we revise this section and include the information to explain how we can conclude from these figures (Page 15, yellow labels; and Page 32, yellow labels).

6. Line 334, 384, 389. Reference to expanding structures and stress fibers. There is no evidence that a myosin-containing structure is expanding. What is more likely happening, is that myosin fluidizes the cortical network, leading to turnover that allows 'cell expansion' through stress dissipation. I would point to other studies that have shown the role of myosin in actin turnover and the role of actin turnover in stress dissipation (PMID: 30538127).

Answer: We thank the reviewer for proposing this interesting view to explain how myosin pulsatile structures facilitate cell expansion process, which looks like to present expanding properties. It seems that Myosin-II pulsatile signals might fluidize the cortical networks and their linked focal adhesions to execute cell expansion through stress dissipation, somehow similar to a reported role of Myosin-II on in vitro F-actin networks (PMID: 30538127). However, whether Myosin-II governs actin turnover or uses other unknown mechanisms to fluidize the cortical networks and enable stress dissipation is unclear. We now update this description in our text (Pages 15-16, yellow labels).

7. Line 363-366: Reference to Supp Fig 11. Just say no to calling out a figure with 9 panels. Explain how what is in the figure supports your statement.

Answer: We thank the reviewer for pointing out this unclear section. Now we include more detailed descriptions of the figures supporting our statement in the results section. We also include some updates about Supple Fig 10 in the results section. Here, we only focus on the strong phenotype from single and combined factor simulations, as well as the deviation between single-factor simulations and actual experimental observation, in the results section. All these changes are updated in (Pages 16-17, yellow labels).

Furthermore, we do include all detailed information about Supplementary Fig 9 (WT simulation), Supplementary Fig 10 (Single-factor simulations) and Supplementary Fig 11 (Combined-factor simulations), in the mathematical modelling section of the Methods (in pages 46-52).

---Additional points from the reviewer's suggestions: "I will give some specific examples, but these examples reflect a broader problem throughout the entire manuscript. It requires careful editing and a logical and consistent ordering of explanations of figure panels for readers to not feel lost."

Answer: As the reviewer suggested, we include more improvement in the following sections:

1) In pages 5-6, specify Supplementary "Fig. 2a-g" into "Supplementary Fig. 2a-e" and "Supplementary Fig. 2f, g" for different descriptions of subcellular changes.

2) In page 6, specify "Supplementary Fig. 2h-o" into "Supplementary Fig. 2h-j): temporally, posterior follicle cells enter the ruffling phase sequentially, but simultaneously enter the increasing and spreading phases; spatially, both ruffling and spreading processes progressively expand from

posterior to anterior (Supplementary Fig. 2j). Notably, ruffling changes and basal actomyosin gradual modulation were detected in posterior wave-occurring cells, but not in anterior cells before wave arrival (Supplementary Fig. 2k-o).”

3) In pages 8-9, specify “Fig. 2d-i” into “Fig. 2d-f” and “Fig. 2g-i” for two different phases; specify “Fig. 2j-o and Supplementary Fig. 4r-t” into “Fig. 2j-l”, “Supplementary Fig. 4r-t” and “Fig. 2m-o” for 3 different phases.

4) In page 10, specify “Supplementary Fig. 5e-j” into “Supplementary Fig. 5e-g” and “Supplementary Fig. 5h-j” for two different optogenetic conclusions.

5) In pages 11-12, specify “Fig. 1i, Fig. 4a, b and Supplementary Fig. 7a, b” into “Supplementary Fig. 7a”, “Fig. 4a and Supplementary Fig. 7b”, “Fig. 1i” and “Fig. 4b” for different cellular changes described in this whole sentence.

6) In page 12, specify “Fig. 4j, k and Supplementary Fig. 7j, k” into “Fig. 4j, k and Supplementary Fig. 7j” and “Supplementary Fig. 7k” for different descriptions of cell behaviors.

Minor comments:

1. Lines 25, 50: ‘junction’ should be ‘junctions’

2. Line 105: ‘1j, j’ should be ‘1i, j’

3. Line 194: ‘blockade of cell spreading process’ should be ‘blockade of the cell spreading process’

Answer: We correct all these as the reviewer suggested (Page 2, 3, 5, 9, yellow labels).

To make it easier for reviewers to track our changes, we mark new insertions in yellow and deletions as strikethroughs.

Reviewer #3 (Remarks to the Author):

The authors have done a commanding job at addressing reviewers' comments. Their revised discussion is overall clearer and more balanced. Regarding their detailed responses, I agree that identifying the RhoGAP acting to dampen RhoA early ones beyond the scope of this manuscript if there are multiple GAPs and GEFs involved, but it is really good that they initiated that search. This said, if the comment on the fact that there might be a need to dampen RhoA activity to initiate the process, they could mention the issue in their discussion towards lines 475. I would also recommend that they include the 1st "Rebuttal Figure 1" as a supplementary figure in the final manuscript, and potentially the 4th Rebuttal Figure. I now fully support publication in Nature Communications.

Answer: We thank the reviewer for improving our manuscript.

- 1) About the inclusion of the 1st "Rebuttal Figure 1", we agree with reviewer 1's comments (regarding that the data is too premature, it is better not to be included in this paper).
- 2) About the inclusion of the 4th "Rebuttal Figure 1", we find it difficult to be in accordance with other supplementary figures and therefore we decide not to include it in this paper.
- 3) About the inclusion of "to dampen RhoA activity to initiate the process", we now include it in the sentence "more susceptible to remodeling of the cytoskeleton and focal adhesions (possibly triggered by the Rho1 down-regulation in follicle cells)", in (Page 21, yellow labels).

REVIEWERS' COMMENTS

Reviewer #2 (Remarks to the Author):

The clarity of the manuscript and the meaning of specific figure panels is now much improved. I commend the authors on their efforts to revise the manuscript and recommend publication.